# SCR²-ST: Combine Single Cell with Spatial Transcriptomics for Efficient Active Sampling via Reinforcement Learning

**Junchao Zhu**[1]      JUNCHAO.ZHU@VANDERBILT.EDU
**Ruining Deng**[2]      RUD4004@MED.CORNELL.EDU
**Junlin Guo**[1]      JUNLIN.GUO@VANDERBILT.EDU
**Tianyuan Yao**[1]      TIANYUAN.YAO@VANDERBILT.EDU
**Chongyu Qu**[1]      CHONGYU.QU@VANDERBILT.EDU
**Juming Xiong**[1]      JUMING.XIONG@VANDERBILT.EDU
**Siqi Lu**[3]      SLU09@WM.EDU
**Zhengyi Lu**[1]      ZHENGYI.LU@VANDERBILT.EDU
**Yanfan Zhu**[1]      YANFAN.ZHU@VANDERBILT.EDU
**Marilyn Lionts**[1]      MARILYN.M.LIONTS@VANDERBILT.EDU
**Yuechen Yang**[1]      YUECHEN.YANG@VANDERBILT.EDU
**Yalin Zheng**[4]      YALIN.ZHENG@LIVERPOOL.AC.UK
**Yu Wang**[5]      YU.WANG.2@VUMC.ORG
**Shilin Zhao**[5]      SHILIN.ZHAO.1@VUMC.ORG
**Haichun Yang**[5]      HAICHUN.YANG@VUMC.ORG
**Yuankai Huo**[1]      YUANKAI.HUO@VANDERBILT.EDU

[1] *Vanderbilt University, TN, USA*

[2] *Weill Cornell Medicine, NY, USA*

[3] *The College of William and Mary, VA, USA*

[4] *University of Liverpool, UK*

[5] *Vanderbilt University Medical Center, TN, USA*

**Editors:** Accepted for publication at MIDL 2026

## Abstract

Spatial transcriptomics (ST) is an emerging technology that enables researchers to investigate the molecular relationships underlying tissue morphology. However, acquiring ST data remains prohibitively expensive, and traditional fixed-grid sampling strategies lead to redundant measurements of morphologically similar or biologically uninformative regions, thus resulting in scarce data that constrain current methods. The well-established single-cell sequencing field, however, could provide rich biological data as an effective auxiliary source to mitigate this limitation. To bridge these gaps, we introduce SCR²-ST, a unified framework that leverages single-cell prior knowledge to guide efficient data acquisition and accurate expression prediction. SCR²-ST integrates a single-cell guided reinforcement learning-based (SCRL) active sampling and a hybrid regression-retrieval prediction network SCR²Net. SCRL combines single-cell foundation model embeddings with spatial density information to construct biologically grounded reward signals, enabling selective acquisition of informative tissue regions under constrained sequencing budgets. SCR²Net then leverages the actively sampled data through a hybrid architecture combining regression-based modeling with retrieval-augmented inference, where a majority cell-type filtering mechanism suppresses noisy matches and retrieved expression profiles serve as soft labels for

auxiliary supervision. We evaluated SCR²-ST on three public ST datasets, demonstrating SOTA performance in both sampling efficiency and prediction accuracy, particularly under low-budget scenarios. Code is publicly available at: https://github.com/hrlblab/SCR2ST.

**Keywords:** Computational Pathology, Spatial Transcriptomics, Active Learning

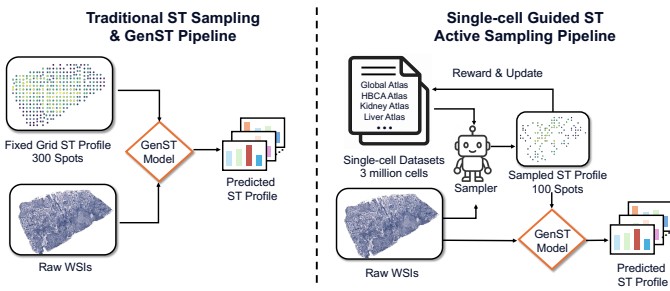

Figure 1: **Comparison between traditional ST sampling and our active sampling.** *Left:* Traditional ST methods rely on fixed-grid sampling regardless of biological importance, leading to redundant measurements in similar regions and inefficient use of sequencing budgets. *Right:* Our proposed approach actively selects informative spots by incorporating single-cell prior knowledge, reducing redundancy while preserving biologically diverse regions.

## 1. Introduction

Spatial transcriptomics (ST) provides a new perspective for studying the relationship between pathological tissue structures and their spatial gene expression patterns (Burgess, 2019; Asp et al., 2019, 2020). However, acquiring ST data remains relatively expensive (Choe et al., 2023), which together pose challenges for large-scale data collection in practice (He et al., 2020; Zhu et al., 2025d).

Histology features exhibit strong correlations with gene expression patterns (Badea and Stǎnescu, 2020), providing a foundation for image-based gene expression prediction (He et al., 2020; Zhu et al., 2025a). Deep learning methods have begun leveraging histology images to infer ST expression profiles of each tissue slide (Xie et al., 2024; Yang et al., 2023; Zhu et al., 2025b,c). Representative approaches include regression-based ST-Net (He et al., 2020), HisToGene (Pang et al., 2021), and EGN (Yang et al., 2023), which directly predict expression values from local image appearance; and retrieval-based vision–omics contrastive learning methods, such as BLEEP (Xie et al., 2024) and mlxSTExp (Min et al., 2024).

However, traditional fixed-grid sampling inevitably acquires many spatially adjacent regions with highly similar morphology, leading to substantial molecular redundancy and reduced biological diversity. Its non-selective nature also results in the inclusion of biologically uninformative areas (Schroeder et al., 2025; Grases and Porta-Pardo, 2025). Consequently, the effective information density of the dataset is low, causing a mismatch between sequencing cost and informative yield, thus constraining the performance and scalability of image-based ST prediction methods.

The limited availability of ST data motivates the integration of external biological knowledge to compensate for inherent constraints in coverage and data quality. In particular, the single-cell sequencing field provides substantially richer priors, supported by large-scale datasets (Regev et al., 2017) and powerful foundation models (Cui et al., 2024), with sample sizes typically exceeding ST by more than an order of magnitude (Svensson et al., 2018). Single-cell profiles resolve cellular types, states, and regulatory programs (Cao et al., 2019; Stuart et al., 2019), offering mechanistic insight into gene expression variation across tissues. Incorporating such fine-grained priors into ST analysis introduces valuable structural guidance and biological constraints, helping mitigate challenges related to limited sampling.

To achieve this, we introduce SCR2-ST, a unified framework that leverages single-cell prior knowledge to guide both efficient data acquisition and expression prediction. Our framework comprises two components. First, we develop a single-cell guided reinforcement learning-based (SCRL) active sampling strategy that jointly leverages single-cell priors and spatial tissue cues to construct a biological reward function, which enables the policy network to adaptively prioritize informative regions while avoiding redundant measurements, maximizing the utility of each sequenced spot under constrained budgets. Within this framework, we further propose a hybrid regression-retrieval prediction network SCR2Net, which integrates regression modeling with retrieval-augmented inference. The retrieval branch aggregates signals from morphologically similar spots, while a majority cell-type filtering mechanism suppresses unreliable matches, balancing global structural learning with context-aware expression transfer. Our contributions can be summarized as fourfold:

- We introduce SCR$^2$-ST, a pioneering and generalizable framework that leverages single-cell prior as an auxiliary source to overcome the scarcity of ST data. It jointly enables efficient data acquisition and accurate expression prediction for ST profile.

- Within this framework, we propose a reinforcement learning-based (SCRL) active sampling strategy that prioritizes informative regions under constrained sequencing budgets through biologically grounded reward signals.

- Building upon this, we develop SCR$^2$Net, a hybrid prediction network that integrates direct regression with retrieved soft label supervision, with a majority cell-type filtering module that suppresses unreliable matches in heterogeneous tissues.

- We provide a systematic benchmark across three public ST datasets under varied sampling budgets, with code and tools released to support reproducible research.

## 2. Method

### 2.1. Overall Framework

We introduce SCR$^2$-ST, a unified framework that leverages single-cell prior knowledge to guide efficient data acquisition and accurate expression prediction under limited data budgets, as illustrated in Figure 2. Within SCR$^2$-ST, our single-cell-guided reinforcement learning-based (SCRL) active sampling strategy integrates single-cell priors with spatial features, using a multi-objective reward function to iteratively drive the policy toward selecting the most informative spots. To fully exploit the abundant single-cell priors in prediction, we

design a hybrid regression-retrieval prediction network $SCR^2Net$ that fuses direct regression with retrieval-augmented soft label supervision on the actively sampled set.

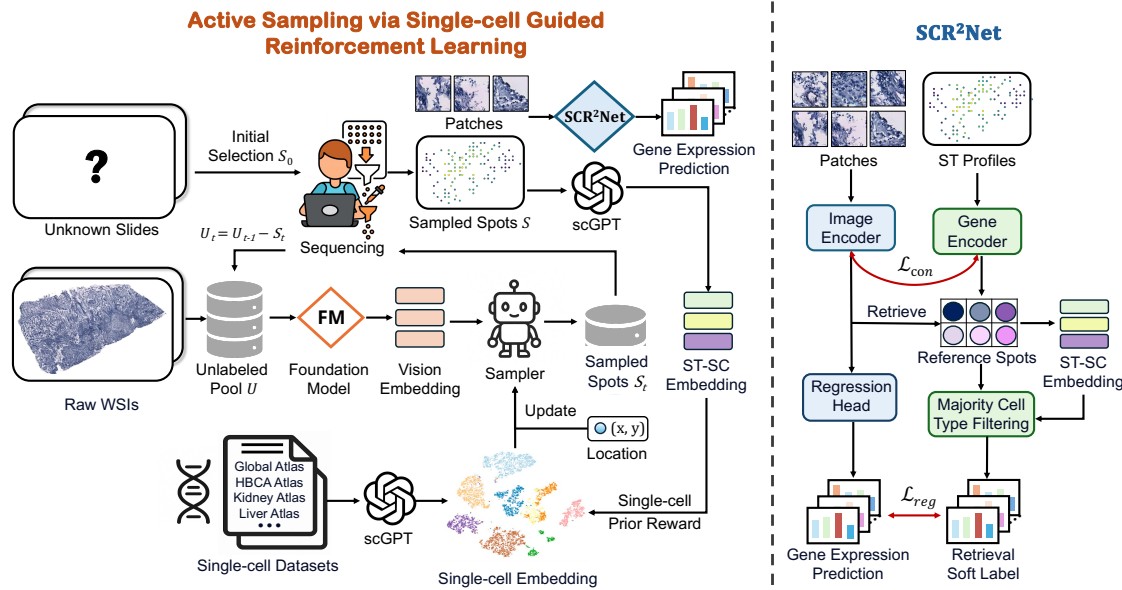

Figure 2: **Overview of our proposed $SCR^2$-ST framework.** *Left*: Single-cell–guided reinforcement learning–based (SCRL) active sampling strategy that integrates vision features and external single-cell priors to iteratively select informative spots, ensuring efficient data acquisition under limited data budgets. *Right*: Our $SCR^2Net$ that infers gene expression from histology images with retrieval-augmented reference. Majority cell-type filtering guided by single-cell priors is applied to suppress unreliable matches in heterogeneous regions.

## 2.2. Active Sampling via Single-cell Guided Reinforcement Learning

### 2.2.1. Policy Network for Active Sampling

Prior to active sampling, we perform dense visual feature extraction on tissue sections. Specifically, we uniformly partition WSIs patches and employ the pre-trained UNI (Chen et al., 2024) to extract visual embeddings $\{e_i\}_{i=1}^N$ for each patch, along with their corresponding spatial coordinates $\{(x_i, y_i, w_i)\}_{i=1}^N$, where $w_i$ denotes the slide identifier. Based on these embeddings, we construct a lightweight policy network $\pi_\theta(\cdot)$ that outputs a sampling priority score for each candidate location as

$$\pi_\theta(e_i) = W_2 \cdot \text{ReLU}(W_1 e_i), \tag{1}$$

where $W_1 \in \mathbb{R}^{128 \times d}$ and $W_2 \in \mathbb{R}^{1 \times 128}$ are learnable parameters. The scores are then normalized into a probability distribution via softmax as $p_i = \frac{\exp(\pi_\theta(e_i))}{\sum_{j=1}^N \exp(\pi_\theta(e_j))}$. At iteration $t$, the policy network samples $k$ new locations $S_t$ from the unsampled candidate set $\mathcal{U}_t$

according to this probability distribution, and adds them to sampling pool $\mathcal{S} = \bigcup_{\tau \leq t} S_\tau$. The sampling process terminates when the number of samples reaches the total budget $B$.

### 2.2.2. Multi-Objective Reward Design

After obtaining sample set $S_t$ at round $t$, we evaluate sampling quality and construct multi-objective reward signals to update the policy network. We extract ST expression embeddings $\{\mathbf{z}_i\}_{i \in S_t}$ by pretrained scGPT (Cui et al., 2023) for sampled locations and reference embeddings $\{\mathbf{q}_j\}_{j=1}^M$ from external single-cell data. The reward function measures sampling quality from two complementary perspectives: biological diversity and spatial uniformity.

**Single-Cell Prior-Guided Biological Diversity Reward.** To quantify how well the sample set explores the single-cell state space, we first apply PCA to reduce the single-cell embeddings $\{\mathbf{q}_j\}$ to 50 dimensions, then cluster them into $C$ latent cell state clusters using MiniBatchKMeans, obtaining the cluster center set $\{\boldsymbol{\mu}_c\}_{c=1}^C$. The coverage reward measures the fraction of clusters reached by the sample set:

$$R_{\text{sc}}(S_t) = \frac{|\{\arg\min_c \|\mathbf{z}_i - \boldsymbol{\mu}_c\|_2 : i \in S_t\}|}{C}, \tag{2}$$

where $|\cdot|$ denotes set cardinality. Each sampled point $\mathbf{z}_i$ is assigned to its nearest cluster, and coverage is computed as the ratio of unique clusters covered to total clusters $C$.

We then match each ST embedding $\mathbf{z}_i$ to the most similar single-cell embedding via cosine similarity and retrieve the corresponding cell type label to evaluate the cell type diversity of selected samples as:

$$j^*(i) = \arg\max_j \frac{\mathbf{z}_i^\top \mathbf{q}_j}{\|\mathbf{z}_i\|\|\mathbf{q}_j\|}, \tag{3}$$

We compute the cell type distribution in the sample set as $P(k) = |\{i \in S_t : \text{type}(j^*(i)) = k\}|/|S_t|$, and define the diversity reward based on normalized entropy:

$$R_{\text{type}}(S_t) = \frac{-\sum_k P(k)\log(P(k) + \varepsilon)}{\log(K + \varepsilon)}, \tag{4}$$

where $k$ indexes individual cell types, $K$ is the number of distinct cell types observed in the sample set, and $\varepsilon$ is a small constant for numerical stability, encouraging preferential sampling of regions with greater cellular heterogeneity.

**Spatial Distribution Diversity Reward.** Spatial distribution of sampled points also affects information density. An ideal sampling strategy should balance two objectives: (1) spatial dispersion to avoid over-clustering in local regions; (2) uniform coverage to ensure that unsampled locations have nearby reference points. Therefore, we define dispersion $D_{\text{disp}}$ as the average pairwise distance among sampled points, where larger values indicate better dispersion. We define coverage $D_{\text{cover}}$ as the average distance from all candidate locations to their nearest sampled point, where smaller values indicate more uniform coverage:

$$D_{\text{disp}}(S_t) = \frac{1}{|S_t|^2} \sum_{i,j \in S_t} \|(x_i, y_i) - (x_j, y_j)\|_2, \quad D_{\text{cover}}(S_t) = \frac{1}{N} \sum_{i=1}^N \min_{j \in S_t} \|(x_i, y_i) - (x_j, y_j)\|_2. \tag{5}$$

The spatial distribution diversity reward combines both metrics:

$$R_{\text{spa}}(S_t) = \frac{D_{\text{disp}}(S_t) + D_{\text{cover}}(S_t)}{2}. \tag{6}$$

### 2.2.3. COMBINED REWARD AND POLICY OPTIMIZATION

We linearly combine the three reward components into a composite signal:

$$R(S_t) = w_{\text{sc}} \cdot R_{\text{sc}}(S_t) + w_{\text{type}} \cdot R_{\text{type}}(S_t) + w_{\text{spa}} \cdot R_{\text{spa}}(S_t), \tag{7}$$

where $w_{\text{sc}}$, $w_{\text{type}}$, and $w_{\text{spa}}$ control the relative contributions of single-cell manifold coverage, cell type diversity, and spatial distribution diversity, respectively. We then update the policy network parameters using the composite reward:

$$\nabla_\theta \mathcal{J} = \mathbb{E}_{S_t \sim \pi_\theta} \left[ R(S_t) \cdot \nabla_\theta \log \pi_\theta(S_t) \right], \tag{8}$$

where $\mathcal{J}$ is the expected cumulative reward. Through gradient ascent optimization, the policy network progressively learns to balance biological diversity and spatial uniformity, steering the sampling strategy toward more informative tissue regions.

## 2.3. SCR$^2$Net: Single-Cell Guided Regression-Retrieval Network

To further leverage single-cell prior knowledge, we design SCR$^2$Net with two complementary paths, including a direct regression path for image-to-expression mapping, and a retrieval-augmented path that provides soft supervision by retrieving similar samples from the training set as an external knowledge base.

### 2.3.1. SINGLE-CELL GUIDED RETRIEVAL MODULE

**Cross-Modality Alignment.** Direct regression alone struggles to capture complex expression patterns under limited training samples. To address this, we introduce a retrieval-augmented module that treats the training set as an external memory bank encoding single-cell knowledge, providing soft supervision for the regression pathway.

We design two projection heads with identical architecture to map image features $f_{img}$ from the visual encoder and gene expression embeddings into a shared semantic space. An InfoNCE loss $\mathcal{L}_{con}$ is applied to align vision-omics representations and update the projection head. We then compute cosine similarity between the query image and reference samples:

$$\text{sim}(f_{img}, y_j) = \frac{\phi_{\text{img}}(f_{img})^\top \phi_{\text{expr}}(y_j)}{\|\phi_{\text{img}}(f_{img})\| \|\phi_{\text{expr}}(y_j)\|}, \tag{9}$$

where $\phi_{\text{img}}$ and $\phi_{\text{expr}}$ denote the image and expression projection heads, respectively. We select the top-$K$ most similar samples to construct the reference set.

**Cell-Type-Aware Filtering and Knowledge Distillation.** Expression patterns vary significantly across cell types in ST data, while vision representation could resemble, thus directly aggregating all retrieved samples may introduce noise. To ensure biological consistency, we introduce a majority cell-type filtering mechanism. We count cell type distribution among the top-$K$ samples and retain only those belonging to the $T$ most frequent cell types.

The mean expression of filtered samples serves as the retrieved soft label $\hat{y}_{\text{ret}} = \frac{1}{|\mathcal{R}|} \sum_{j \in \mathcal{R}} y_j$, where $\mathcal{R}$ denotes the filtered retrieval set.

To account for retrieval quality, we introduce a similarity-based confidence mask $m$, where higher retrieval similarity leads to greater weight on the distillation loss. The retrieved prediction $\hat{y}_{\text{ret}}$ then guides the regression path through knowledge distillation, with loss function $\mathcal{L}_{ret}$ weighted by a hyperparameter $\lambda_{\text{KD}}$ denoted as:

$$\mathcal{L}_{ret} = \lambda_{\text{KD}} \cdot m \cdot \|\hat{y} - \hat{y}_{\text{ret}}\|^2 \tag{10}$$

### 2.3.2. REGRESSION PATH AND TRAINING OBJECTIVE

We adopt DenseNet-121 (Huang et al., 2017) pre-trained on ImageNet as the visual encoder to capture histomorphological patterns. Given an input patch, the encoder produces a compact feature vector through global average pooling. A two-layer MLP then decodes the visual features into gene expression predictions $\hat{y}$, forming the direct regression path.

To supervise the regression prediction, we employ two complementary losses. The MSE loss $\mathcal{L}_{\text{reg}} = \|y - \hat{y}\|^2$ directly minimizes the difference between predictions and ground truth, while a Pearson Correlation Coefficient (PCC) loss $\mathcal{L}_{\text{pcc}} = 1 - \text{PCC}(y, \hat{y})$ to capture the correlation structure across genes, which is important for preserving gene-spatial relationships. The total loss integrates direct supervision from both regression losses and soft supervision from the retrieval-based distillation with hyperparameters $\lambda_r$ and $\lambda_p$, denoted as:

$$\mathcal{L} = \lambda_r \cdot \mathcal{L}_{\text{reg}} + \lambda_p \cdot \mathcal{L}_{\text{pcc}} + \mathcal{L}_{\text{ret}}, \tag{11}$$

## 3. Data and Experiments

**Datasets and Preprocessing.** We evaluate all methods using three public ST datasets, including HER2 (Andersson et al., 2021), Breast Cancer (He et al., 2020), and Kidney (Lake et al., 2023). For each spot, we cropped a $224 \times 224$ pixel patch centered on spatial coordinates as model input. We selected top 300 genes with highest average variance as prediction targets. Following BLEEP (Xie et al., 2024), we applied a $\log(1 + x)$ transformation on raw readouts. For the external single-cell datasets, we use two million cells from (Lake et al., 2025) as the reference for the Kidney dataset, and around three million cells from (Chen et al., 2025; Reed et al., 2024; Klughammer et al., 2024) as the reference for the Breast Cancer and HER2 datasets. A detailed profile of datasets is provided in Appendix A.

**Baseline.** We compared our model against SOTA methods, including regression-based models ST-Net (He et al., 2020), EGN (Yang et al., 2023), HisToGene (Pang et al., 2021), His2ST (Zeng et al., 2022), and TRIPLEX (Chung et al., 2024), and retrieval-based models BLEEP (Xie et al., 2024) and mclSTExp (Min et al., 2024). All methods were trained and evaluated under consistent experimental settings to ensure fair comparison. To validate our sampling strategy, we select Monte Carlo random sampling, uncertainty-based sampling (Safaei et al., 2024), and diversity-driven sampling (Zhdanov, 2019) for comparison.

**Evaluation Metrics.** We employed Pearson correlation coefficient (PCC), mean squared error (MSE), and mean absolute error (MAE) to comprehensively assess model performance in gene expression prediction from both spatial correlation and error perspectives.

**Implementation Details.** All experiments were conducted on a single NVIDIA RTX A6000 GPU. We employed SGD optimizer with momentum of 0.9 and weight decay of

$10^{-4}$. The initial learning rate was set to $lr_0 = 10^{-4}$, with a cosine annealing schedule that gradually decays the learning rate to $10^{-6}$. The training batch size was set to 256. Details of experimental implementation and hyperparameter settings are listed in Appendix B.

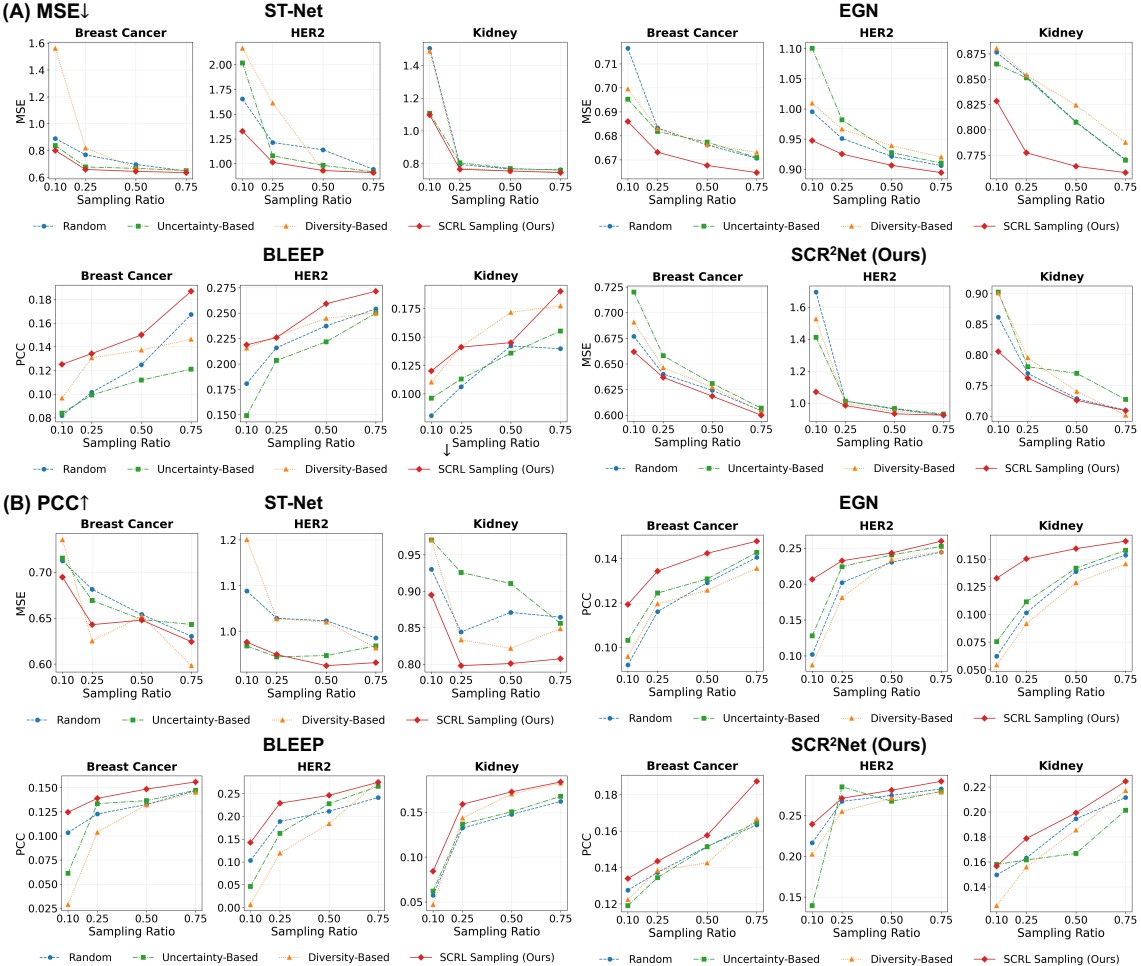

Figure 3: **Comparison of sampling strategies across multiple methods with metrics reported (A) MSE and (B) PCC.** We evaluate performance under data ratios of 10%–75% to simulate low-resource scenarios, with each curve denoting a sampling strategy. Our SCRL sampling consistently achieves better performance across datasets and models, particularly in low-budget (10%–25%) regimes.

## 4. Results

### 4.1. Active Sampling under Budget Constraints

To validate the effectiveness of our active sampling strategy via single-cell guided reinforcement learning (SCRL), we conducted a systematic evaluation across methods. As shown in

Figure 3 and Figure 7 in Appendix C, we compared four sampling strategies under training data ratios ranging from 10% to 75%. Experimental results demonstrate that SCRL sampling achieves optimal performance across all datasets and model combinations, with particularly advantages in low-budget scenarios (10%–25%). For Breast Cancer dataset at a 10% sampling ratio, SCRL sampling reduces the MSE from approximately 0.85 to 0.75 and improves the PCC from 0.04 to 0.14 on ST-Net compared to random sampling. This trend holds consistently across other datasets and diverse methods.

Notably, we observed differential sensitivity to sampling strategies across model types. Retrieval-based models exhibit greater sensitivity to data quality compared to end-to-end regression models, resulting in larger performance gaps between different sampling strategies. This can be attributed to the nature of contrastive learning, which is highly dependent on the quality and diversity of training samples. Our SCRL sampling strategy balances biological quality and diversity. Specifically, single-cell references ensure that sampled spots cover critical cell subpopulations, while spatial density information guides the sampling process to preserve morphological diversity. This dual constraint enables SCRL to achieve stable and consistent performance improvements across both training paradigms.

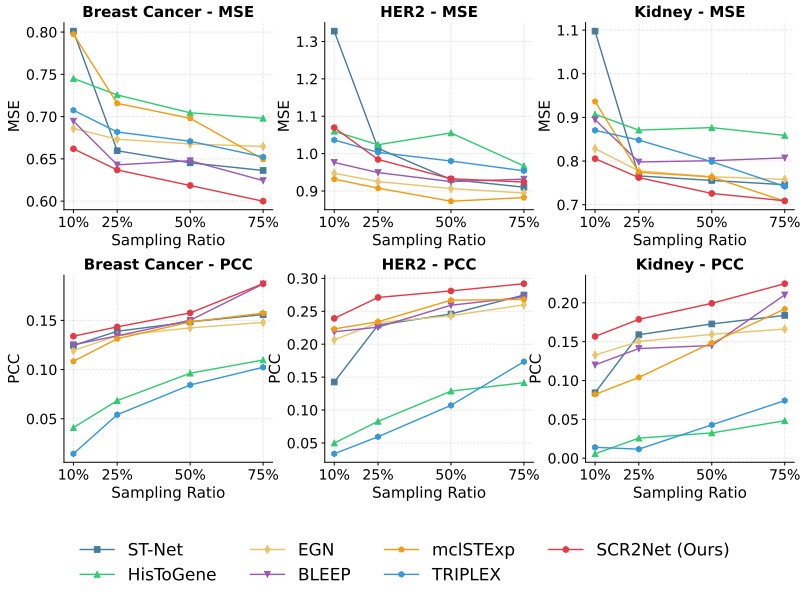

Figure 4: **Comparison of different sampling ratios (10-75%).** Our SCR²Net consistently achieves better metrics, especially under low sampling budget scenarios.

## 4.2. Empirical Validation on Gene Expression Prediction

We conducted four-fold cross-validation at the sample level to validate SCR²Net against SOTAs. Table 1 summarizes quantitative comparisons across different cohorts. Our SCR²Net outperforms existing methods in almost all metrics, achieving the lowest MSE and MAE ↓ as well as the highest PCC on most datasets. For example, on the Kidney dataset, SCR²Net

Table 1: **Performance comparison on gene expression prediction task.** The best performance is highlighted in **orange** and second highest in blue, where we can observe that SCR$^2$Net outperforms the SOTAs across most metrics on most datasets.

| Model | Breast Cancer | | | HER2 | | | Kidney | | |
|---|---|---|---|---|---|---|---|---|---|
| | MSE ↓ | MAE ↓ | PCC ↑ | MSE ↓ | MAE ↓ | PCC ↑ | MSE ↓ | MAE ↓ | PCC ↑ |
| ST-Net | 0.6318 | 0.6377 | 0.1592 | 0.9237 | 0.7559 | 0.2709 | 0.7460 | 0.6811 | 0.1851 |
| His2ST | 0.6999 | 0.6682 | 0.0612 | 0.9928 | 0.8034 | 0.1045 | 0.7912 | 0.7080 | 0.0571 |
| HisToGene | 0.6521 | 0.6486 | 0.1149 | 0.9702 | 0.8050 | 0.1392 | 0.8540 | 0.7373 | 0.1134 |
| EGN | 0.6662 | 0.6558 | 0.1462 | 0.8916 | 0.7640 | 0.2524 | 0.7574 | 0.6864 | 0.1632 |
| TRIPLEX | 0.6672 | 0.6590 | 0.1093 | 0.9356 | 0.7752 | 0.2167 | 0.7168 | 0.6692 | 0.0930 |
| BLEEP | 0.6266 | 0.6044 | **0.2041** | 0.9507 | 0.7613 | 0.2834 | 0.8246 | 0.7167 | 0.2020 |
| mc1STExp | 0.6472 | 0.6202 | 0.1645 | **0.8882** | 0.7367 | 0.2651 | 0.7438 | 0.6759 | 0.1580 |
| SCR$^2$Net (Ours) | **0.5848** | **0.5725** | 0.1940 | 0.9139 | **0.7042** | **0.3028** | **0.7038** | **0.6611** | **0.2391** |

improves PCC by a clear margin to 0.2391, compared with prior baselines of 0.2020. Furthermore, as illustrated in Figure 4 in Appendix C, SCR$^2$Net maintains strong predictive robustness under varying sampling budgets with our SCRL sampling strategy. The performance gap is pronounced under low sampling ratios (10–25%), where other approaches suffer degradation due to sparse tissue coverage. In contrast, SCR$^2$Net mitigates this by acquiring informative tissue regions and leveraging retrieval-based auxiliary priors, resulting in a more reliable predictive performance with reduced sequencing costs.

### 4.3. Ablation Study

**Reward Function in SCRL sampling.** We conducted an ablation analysis on the Biological Prior Reward and the Spatial Density Reward. As shown in Figure 8 in Appendix C, when using only Biological Reward, model performs well at low sampling ratios (10% and 25%); however, as the ratio increases, redundant samples limit further performance gains. Conversely, using only Spatial Reward yields results similar to random sampling, as it cannot directly assess the informativeness of samples. Combining both rewards ensures both biological quality and diversity and allows the model to achieve optimal performance with a performance curve exhibiting a stable upward trend as the sampling ratio increases.

Table 2: **Stability of random initializations.** We reported the performance comparison of our SCRL and random sampling under different random seeds, indicating that our sampling strategy achieves stable performance across different initializations.

| Model | Ratio | SCRL (Ours) | | | Random | | |
|---|---|---|---|---|---|---|---|
| | | MSE ↓ | MAE ↓ | PCC ↑ | MSE ↓ | MAE ↓ | PCC ↑ |
| ST-Net | 0.10 | 0.748±0.049 | 0.688±0.014 | 0.117±0.010 | 0.905±0.111 | 0.754±0.042 | 0.097±0.018 |
| | 0.25 | 0.681±0.032 | 0.667±0.014 | 0.130±0.009 | 0.720±0.046 | 0.674±0.018 | 0.126±0.010 |
| | 0.50 | 0.661±0.012 | 0.643±0.006 | 0.140±0.008 | 0.694±0.035 | 0.673±0.018 | 0.134±0.007 |
| SCR$^2$Net | 0.10 | 0.679±0.015 | 0.659±0.016 | 0.132±0.003 | 0.709±0.037 | 0.672±0.021 | 0.119±0.007 |
| | 0.25 | 0.642±0.011 | 0.642±0.017 | 0.139±0.007 | 0.658±0.027 | 0.660±0.018 | 0.131±0.009 |
| | 0.50 | 0.617±0.009 | 0.629±0.006 | 0.165±0.010 | 0.631±0.015 | 0.639±0.008 | 0.154±0.009 |

**Functional Blocks in SCR²Net.** As shown in Table 3, removing Retrieval Reference Module leads to an increase in MSE from 0.7038 to 0.7460 and a decrease in PCC ↑ from 0.2391 to 0.1851 on the Kidney dataset, which indicates its effectiveness in providing reference priors by incorporating similar spots. Meanwhile, majority cell type filtering mechanism suppresses interference from noisy references by excluding low-quality retrieved spots.

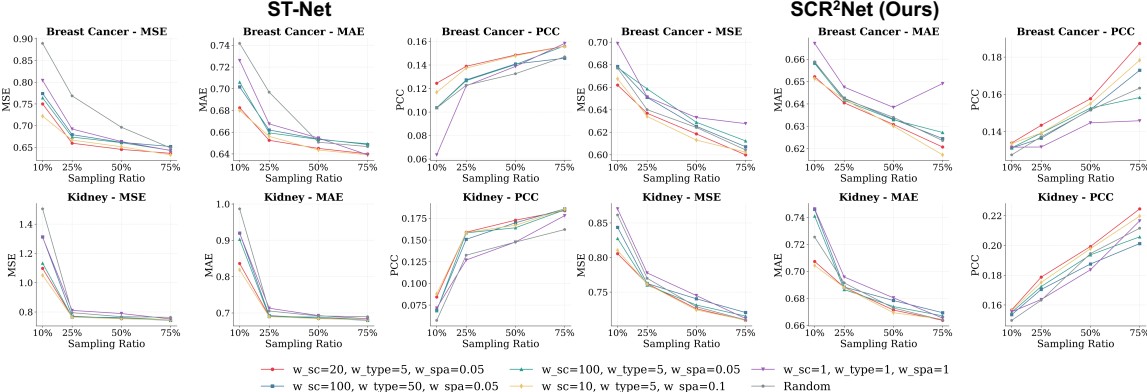

Figure 5: **Sensitivity analysis of reward weight configurations under different sampling ratios.** Each curve represents a distinct reward weight setting, with random sampling shown as a baseline. When reward terms are placed at comparable scales, both models exhibit consistent performance trends as the sampling ratio increases. In contrast, configurations dominated by a single reward term result in clear performance degradation, highlighting the necessity of balancing biological diversity and spatial coverage.

**Sensitivity Analysis of Hyperparameters.** We tested different combinations of candidate pool size $K$, retained cell types $T$, and confidence mask threshold $m$ for the retrieval module. Results in Table 3 indicate that overly small values of $K$ and $T$ (e.g., $K = 10$, $T = 3$) or a higher threshold $m$ limit the richness of reference information and reduce the number of effective reference spots. Conversely, overly large settings of $K$ and $T$ or a lower threshold fail to effectively filter noisy matches, leading to performance degradation. Therefore, moderate hyperparameter settings achieve the optimal trade-off between suppressing noisy matches and preserving informative retrieval references.

**Sensitivity analysis of reward weight.** We conduct a sensitivity analysis of the reward weight configurations by systematically controlling the weights of different reward terms at different scales to evaluate how scale differences affect sampling behavior and downstream prediction performance. Specifically, we construct multiple weight configurations, where some place all reward terms within a comparable scale (e.g., the original setting and nearby configurations), while others assign certain reward terms to significantly different orders of magnitude. As shown in Figure 5 and Figure 8, only when the reward terms are adjusted to comparable scales does the model maintain good and consistent performance; when a single reward term dominates or random sampling is adopted, the performance degrades noticeably. These results suggest that the effectiveness of the proposed reward design does

Table 3: **Ablation study and hyperparameter sensitivity analysis in SCR$^2$Net,** where SCR²Net achieves the optimal results with all blocks, and a moderate hyperparameter setting provides the best balance between noise and information.

| Functional Block & Setting | | Breast Cancer | | | Kidney | | |
|---|---|---|---|---|---|---|---|
| | | MSE ↓ | MAE ↓ | PCC ↑ | MSE ↓ | MAE ↓ | PCC ↑ |
| w.o. Retrieval Reference Module | | 0.6318 | 0.6377 | 0.1592 | 0.7460 | 0.6811 | 0.1851 |
| w.o. Cell Type Filtering | | 0.6032 | 0.5992 | 0.1786 | 0.6952 | 0.6531 | 0.2235 |
| w. All functional blocks | | **0.5848** | **0.5725** | **0.1940** | 0.7038 | **0.6611** | **0.2391** |
| Retrieval Module | $K = 10, T = 3$ | 0.6079 | 0.6198 | 0.1680 | 0.7236 | 0.6814 | 0.2187 |
| | $K = 20, T = 5$ | 0.5912 | 0.6059 | 0.1711 | **0.6886** | 0.6643 | 0.2225 |
| | $K = 100, T = 20$ | **0.5848** | 0.5925 | 0.1839 | 0.7239 | 0.6712 | 0.1977 |
| Confidence Mask | m = 0.05 | 0.6052 | 0.6035 | 0.1743 | 0.7135 | 0.6659 | 0.2102 |
| | m = 0.35 | 0.6233 | 0.6281 | 0.1590 | 0.7465 | 0.7005 | 0.1920 |

not rely on precise weight tuning, but rather on maintaining a balanced contribution between biological diversity and spatial coverage.

**Stability under random initializations.** We evaluate the stability of the proposed adaptive sampling strategy by repeating experiments with different random seeds under multiple sampling ratios. As shown in Table 2, SCRL consistently achieves stable performance with small variance across runs and outperforms random sampling, indicating that the learned sampling behavior is robust to random initialization.

**Limitations and Challenges.** Despite these gains, overall accuracy remains bounded by intrinsic challenges of spatial transcriptomics rather than the sampling strategy alone. The most challenging regions for both sampling and prediction are typically highly heterogeneous or transitional tissue areas, as well as regions containing rare cell populations, which are often clinically important. In addition, morphologically similar regions may exhibit distinct molecular profiles, fundamentally limiting the predictive power of image-based models. Technical noise and spot-level signal averaging inherent to ST measurements further introduce uncertainty.

## 5. Conclusion

We propose SCR²-ST, a unified framework that bridges single-cell prior knowledge with ST to enable efficient data acquisition and accurate expression prediction. Moving beyond traditional fixed-grid sampling, SCRL constructs biologically grounded reward signals by integrating single-cell prior knowledge with spatial density cues, guiding the policy network to prioritize informative regions while avoiding redundant measurements. SCR²Net then fuses direct regression with retrieval-augmented inference, where cell-type-aware filtering suppresses noise and retrieved soft labels regularize the learning process. Extensive experiments on public datasets validate our framework's superior performance across diverse prediction architectures, with notable gains under low-budget constraints. By unifying active sampling and hybrid prediction within a single-cell-empowered paradigm, SCR²-ST offers a scalable solution for efficient spatial transcriptomics modeling and establishes a foundation for future research in budget-aware biomedical data acquisition.

## Acknowledgments

This research was supported by NIH R01DK135597 (Huo), DoD HT9425-23-1-0003 (HCY), NSF 2434229 (Huo), and KPMP Glue Grant. This work was also supported by Vanderbilt Seed Success Grant, Vanderbilt Discovery Grant, and VISE Seed Grant. This project was supported by The Leona M. and Harry B. Helmsley Charitable Trust grant G-1903-03793 and G-2103-05128. This research was also supported by NIH grants R01EB033385, R01DK132338, REB017230, R01MH125931, and NSF 2040462. We extend gratitude to NVIDIA for their support by means of the NVIDIA hardware grant. This work was also supported by NSF NAIRR Pilot Award NAIRR240055.

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

## Appendix A. Additional Details of Dataset

We evaluate all methods on three public spatial transcriptomics (ST) datasets: HER2 (Andersson et al., 2021), Breast Cancer (He et al., 2020), and Kidney (Lake et al., 2023). The HER2 dataset comprises 8 patient samples with 36 WSIs and 13,620 spatial spots in total. The Breast Cancer dataset consists of 23 samples with 68 WSIs and 30,066 spots. The Kidney dataset includes 22 samples with 23 WSIs and 25,944 spots. The spot diameter for HER2 and Breast Cancer is 100 $\mu$m, whereas the Kidney dataset adopts a smaller 55 $\mu$m spot diameter. For the external single-cell datasets, we use two million cells from (Lake et al., 2025) as the reference for the Kidney dataset, and around three million cells from (Chen et al., 2025; Reed et al., 2024; Klughammer et al., 2024) as the reference for the Breast Cancer and HER2 datasets.

For each spot location, we extracted a $224 \times 224$ pixel histology patch centered on its spatial coordinate as model input. To construct the prediction targets, we selected the top 300 genes with the highest variance in expression within each dataset. Following BLEEP (Xie et al., 2024), we applied a $\log(1+x)$ transformation to the raw count matrices to alleviate the heavy-tailed distribution characteristic of ST expression data (He et al., 2020). The dataset-specific selected genes are visualized in Appendix Figure 6.

| Dataset | Selected Genes |
|---|---|
| **HER2** | 'IGKC', 'I1GHG3', IGLC2', 'MUCL1; IGHM', 'IGHAI, 'ERBB2', IGLC3'; 'CALMLS', IGHG4', PTPRF, 'FNI, 'ACTGI, 'PSMB3', 'MIEN1, 'TPT1, 'S100A9, 'CISD3', 'BEST! 'COL1A2', 'IGHG1', 'FASN', 'COLIAI', 'ACTB', 'B2M', 'TFF3', 'GRB7', 'APOE, 'FTHI', 'PCGF2', 'MGP', 'PABPCI', 'SERF2', 'PSMD3', 'COL3A1', 'S100A8', 'DDXS', 'NDUFB9', 'HLA-B', 'S100A6', 'GNAS', 'TMSB10', 'EEF2', 'ADAM15', 'MLLT6', 'KRT81', 'AZGPI', 'TRAF4', 'TUBA1B', 'CD24', 'LASPI', 'S100A14', 'SCD', 'KRT7', 'STARD3'', 'TAGLN2', 'PERP', 'AEBP!', 'CD74', 'PTMS', 'HLA-DRA', 'CRIP2', 'IFI27', 'PPPIRIB', 'VMP1', 'PFNI', FTL', 'COX7C', 'GNB2', 'EEFID', 'MYL6, 'P4HB', 'FNBPIL', 'GAPDH', 'FAU', 'SPINT2', 'PRDXI', 'ATPIAI', 'SYNGR2', 'XBP1', 'CCT3'", 'MMP14', 'HNRNPA2B1', 'COX6C", 'ORMDL3', 'EIF4G1', 'CALR', 'PSAP', 'KRT18', 'PLXNB2', 'CTSD', 'SPARC', 'SSR4', 'HLA-C', 'CLDN3', 'GRINA', 'KRT19', 'APOCI', 'SLC25A6', 'HLA-A', 'EIF4G2', 'RACKI', 'HSP90ABI', 'TAGLN', 'FLNA', 'VIM', 'CIQA', 'ATG10', 'HSP90AAI', 'S100A11', 'BGN, 'HSPB1', 'HLA-E', 'C3', 'CNN3', 'ATP6VOB', 'PRRC2A', 'SPDEF, 'UQCRQ', 'MDK', 'COMP', 'SDCI', 'LMNA', 'POSTN', 'CD63', 'JTB', 'CHCHD2', 'HSPA8, IGFBP2', 'OAZI, 'TSPO', '1DH2', 'MUCI', 'LLGL', 'PGAP3', 'CD99', 'PRSS8', 'MIDN', 'CIBI, 'TUBB', 'A2M', 'ALDOA', 'SCANDI, 'COPS9', 'MZT2B', 'LUM', 'LY6E', 'LGALS3', 'CLDN4', 'ELOVLI', 'CST3', 'CFLI', 'CALM', 'NACA', 'COL6A2', 'PSMB4', 'DBI, 'LAPTM4A', 'GPX4', 'PIP4K2B', 'ERGICI', 'SUPT6H', 'NUCKSI, 'SF3B5', 'DHCR24', 'TXNIP', 'MYL9', 'PEBPI', 'LAPTMS', 'UBA52', 'HSP90B1', 'SEC61A1', 'MMACHC", 'MAP3K12', 'MAPKAPK2', 'PNMT', 'CTSB', 'NBLI', 'DDIT4', 'FUCA2', 'JUP', 'PLD3', 'BSG', 'NUPRI', 'LGALSI', 'LGALS3BP', 'UBC', 'SNRPB', 'PTMA', 'BST2', 'PCSK7', 'ISG15', 'KDELRI', 'IGFBP7', 'COX6B1', 'UBE2M', 'COX4II', 'PPPICA', 'COL18A1', 'COPE', 'CYBA', 'PTGES3', 'MYH9', 'TAPBP', 'CD81', 'EDF1', 'SSR2', 'CLTC, 'HEBP', 'RALY', 'PPDPF', 'ZYX', 'HINTI', 'INTS1', 'PCBP2', 'TMBIM6', 'ZBTB7B', 'COX5B', 'ACTN4', 'ZFP36L1', 'FADS2', 'CTTN', 'TIMP1', 'LMAN2', 'STARD10', 'VCP', 'CHPF', 'C12ORF57', 'UBL5', 'GUKI', 'DCN', 'PFDN5', 'SLC2A4RG', 'C1QB', 'S100A10', 'ENO1', 'TMSB4X', 'KRT8', 'SREBF1', 'CLDN', 'PKM', 'SRRM2', 'ARHGDIA', 'IFI6', 'NDUFB7', 'ABCD3', 'PPP1R14B', 'ROMOL', 'FKBP2', 'PFKL', 'KDELR2', 'NDUFA3', 'MRPL12', 'LSM7', 'CIORF122', 'CPD', 'SMARCD2', 'NDUFA4', 'LDHB', 'RABACI', 'ATP6AP1', 'ITM2B', 'SEC61G', 'SERINC2', 'RRBP1', 'EPCAM', 'PGK 1', 'HLA-DPAI', 'HM13', 'TMED9', 'UQCR11', 'CYB561', 'TSTDI', 'PTBPI', 'LPCATI', 'TRIM28', 'UBB', 'AP2S1', 'SH3BGRL3', 'EIF3B', 'AKTI', 'PSMD8', 'SDF4', 'CCDC152', 'MCLI', 'PSMB1', 'UQCR10', 'LSM4', 'EIF4EBP1', 'CD46', 'PDIA6', 'SODI', 'APP', 'TYMP', 'AP1S1', 'CSDE1', 'ARPCIB', 'CD9', 'POLRL', 'CCNDI' |
| **Kidney** | IGKC, 'UMOD', 'MT-CO1', 'GPX3', 'SPPI', 'MT-CO2', 'DEFB1', 'IGHG3', 'IGHG4', 'ALDOB', 'ATP6VOC', 'IGLC2', 'IGHG1', 'IGHAI', 'RPS23', 'MT-CO3', 'RPS12', 'MT-ND3', 'RPLPI', 'RPS27', 'RPS18', 'RPL34', 'RPS15A', 'MT-NDI', 'RPL37A', 'MT-CYB', 'SLC12AI', 'MTIG', 'MT-ATP6', 'RPS27A', 'RPL37', 'RPL41', 'IGFBP7', 'APOE', 'RPS15', 'RPL12', 'RPL17', 'EEFIAI', 'RPL26', 'RPS21', 'RPS24', 'ATPIAI', 'RPS28', 'TMSB4X', 'WFDC2', 'RPL32', 'RPL30', 'PCKI', 'RPS4X', 'RPS8', 'RPS14', 'RPLI1', 'RPS13', 'C7, 'MT-ND2', 'RPL19', 'RPS7', 'A2M', 'RPS29', 'RPS25', 'MGP', 'RPS16', 'RPS6', 'RPS2', 'MT-ND4', 'RPL9', 'ATPIBI', 'RPL7A', 'FAU', 'RPL35A', 'IGLC3', 'PDZK11P1', 'RPL28', 'TPTI', 'RPL10', 'PIGR', 'RPLP2', 'RPL15', 'MT-ND5', 'FTH1', 'COX7C', 'RPS3A', 'RPL23A', 'HLA-C', 'RPS19', 'HSD11B2', 'TMSB10', 'GATM', 'RPL5', 'RPL38', 'ATPSF1E', 'RPS26', 'RPL31', 'SERF2', 'UQCRQ', 'KNG1', 'AQP2', 'SOD2', 'TAGLN', 'RPL24', 'RPL35', 'RPL13A', 'PTHIR', 'RPL21', 'PPIA', 'S100A6', 'RPS20', 'RPS9', 'RPL3', 'RPLPO', 'RPL18', 'RPS10', 'AQPI', 'RPL36A', 'NDUFA4', 'COX4II', 'SPARC', 'HLA-DRA', 'CST3', 'VIM', 'UQCR11', 'RPS3', 'RPL10A', 'DCN', 'CD24', 'COX5B', 'CD74', 'RPL29', 'RPL6', 'SERPINAI', 'RPS17', 'CKB', 'GAPDH', 'SLC12A3', 'COX6AI', 'IGLC1', 'FTL', 'B2M', 'RPL8', 'COX6BI', 'MALATI', 'CTSB', 'RPL18A', 'RPL22', 'GSTPI', 'COX8A', 'RPS11', 'ATPSMG', 'UQCRB', 'HLA-A', 'CHCHD10', 'RPL7', 'HLA-B', 'TTM2B', 'ACTGI', 'IGFBPS', 'LDHB', 'COL1A2', 'CDH16', 'RPL23', 'FXYD2', 'PODXL', 'RPL13', 'RPL14', 'CD81', 'NDUFB2', 'S100A11', 'CALBI', 'IFITM3', 'SELENOP', 'EEFIG', 'RPL27', 'RPL36', 'COX6C", 'ATP5MC3', 'ADIRF', 'MUCI', 'PTGDS', 'GNAS', 'CLCNKB', 'COX7B', 'CTSD', 'MYL9', 'MT2A', 'MAL', 'RACKI', 'ACTB', 'MT-ND4L', 'CA12', 'NAT8', 'CRYAB', 'ADGRG1', 'COX7A2', 'PEBPI', 'EIFI', 'PSAP, 'UBA52', 'REN', 'ATPSF1B', 'NDUFAI', 'ATPSME', 'RPSA', 'TIMP3', 'NEATI', 'ATP5F1A', 'UBLS', 'MTIE', 'NACA', 'MMP7', 'COL18A1', 'CD63', 'TIMP1', 'HINT1', 'ENO1', 'ELOB', 'UQCRH', 'UBB', 'RPL27A', 'RPS5', 'MIF, 'MYL6', 'AEBPI', 'TOMM7', 'ATPSPF', 'ACTA2', 'SLC25A6', 'CLU', 'SLC25A5', 'SRP14', 'GPX4', 'S100A10', 'MIOX', 'BSG', 'NDUFBI', 'CTSH', 'PFDNS', 'SLC3AI', 'S100A2', 'PKM', 'TSPANI', 'LUM', 'RNASEI', 'ASSI', 'APP', 'BGN', 'HSPBI', 'RPL4', 'COL3AI', 'TMA7, 'SLC5A12', 'SERPINAS', 'TFITM2', 'SERPING1', 'SPINK1', 'UQCR10", 'PFNI', 'CHCHD2', 'UBC', 'CXCL14', 'TMBIM6', 'SATI', 'LRP2', 'APLP2', 'CFL1', 'SLC13A3', 'EEF2', 'CD59', 'HLA-DRBI', 'CALR', 'PPPIRIA', 'NME2', 'TPII', 'ATP6VOB', 'GSTM3', 'COXSA', 'ANXA2', 'TXNIP', 'PRDX5', 'NDUFB7', 'ATPSF1D', 'OST4', 'PRDXI', 'ATPSMC2', 'SLC25A3', 'CYBSA', 'MYL12B', 'OAZI', 'NDUFB9', 'ATPSPO', 'CYSTMI', 'TUBA1B', 'ANPEP', 'HSP9OAAI', 'FABPI', 'PPDPF', 'IGFBP4', 'DSTN', 'EPASI', 'RNASEK', 'PTMA' |
| **Breast Cancer** | 'TFF3', 'IGLLS', 'ERBB2', 'MUCI', 'MGP', 'RPS3', 'S100A9', 'KRT19', 'HAJ, 'NHERFI', 'COX6C', 'AZGPI, 'TMSB10, 'TF127, 'RPL18A', 'B2M', 'C4B', 'FASN', 'RPL38', 'RPL19', 'APOE', 'HLA-B', 'HSPBI', 'ACTB', 'PABPCI, 'RPS15', 'RPS12', 'FNI', 'IGFBP2', 'RPS8', 'HLA-C', 'FTHI, 'CCNDI, 'ACTA8', 'CTSD', 'RPL31', 'RPL37A', 'EEF2', 'KRT18', 'XBPI, 'TAGLN', 'RPS17', 'RPS21', 'COLIAL', 'BESTI', 'SNHG25', 'ACTGI', 'RPL18', 'RPS18', 'RPS2', 'CST3', 'HLA-A', 'PSMD3', 'UBA52', 'SSR4', 'RPL29', 'RPL13', 'A2M', 'RPS6', 'AEBPI, 'GAPDH', 'RPS4X', 'RPL30', 'FTL, 'RPS11', 'RPLPI', 'RPL8', 'RPS24', 'RPS28', 'RPL28', 'RPS27A', 'RPL10, 'RPS19', 'BST2', 'RPL37, 'RPS27' 'S100A11', 'CD74', 'CLDN3', 'EVL', 'KRT8', 'RPLA', 'RPS14', 'RPL35', 'PPPICA', 'P4HB', 'SYNGR2', 'HSP9OABI', 'HLA-DRA', 'SLC39A6', 'RPLPO', 'RPL27A', 'RPL23', 'RPS29', 'SERF2', 'RPL12', 'LYGE', 'RPSS', 'RACKI', 'GNAS', 'LGALS3BP", 'S100A14', 'MZT2B', 'PFNI, 'IFITM3', 'RPL11', 'HLA-E', 'RPL35A', 'H3-3B', 'RPL9', 'RPL13A', 'MIENI', 'PRDX2', 'GPX4', 'RPL15', 'APOCI', 'BGN', 'CLU', 'PSAP", 'ATP6VOB', 'CIQA', 'ISG15', 'TPTI', 'RPL36', 'IF16', 'RPL23A', 'RPL27', 'RPS9', 'FXYD3'', 'C3', 'GATA3', 'KRT7', 'BSG', 'MZT2A', 'PTMA', 'PLXNB2', 'EIF4A1', 'STARD1O', 'IGFBP', 'RPL3', 'TAPBP', 'RPS16', 'SPDEF', 'NDUFA13', 'ELF3', 'RPS3A', 'ALDOA', 'TAGLN2', 'CD81', 'TMSB4X', 'ATP6AP1', 'COLIA2', 'BCAP31', 'MALATT', 'DDX', 'TUBAIB', 'CFLI', 'MYL9', 'ATPIAI', 'PTPRF', 'RPS23', 'RPL', 'ANPEP', 'CALR', 'RPL32', 'NBEALI', 'FAU' 'GNB2', 'PRSS8', 'RPLP2', 'S100A6', 'RPS13', 'SDCI, 'RPS7', 'UQCRQ', 'SLC25A6', 'PRDXI, 'VIM', 'RPL14', 'GASS', 'RPL4', 'TUFM', 'POLRL', 'NR2F6', 'RPL34', 'TMEM123', 'GRINA', 'UBC', 'H1-10', 'TER3', 'OAZI', 'METRN', 'IGFBP4', 'RPS15A', 'ENOL', 'CISD3', 'PSMB3', 'POSTN', 'RPL7, 'TIMP1', 'FLNA', 'CD24', 'RPS25', 'CRIP2', 'LAPTMS', 'NDUFB9', 'ARHGDIA', 'JUP', 'GSTPI', 'CLDN4', 'COLAI, 'CYBA', 'EDFI', 'RPL24', 'EIF3C', 'NDUFA11', 'COX8A', 'NPDCI', 'SPARC', 'RPS20', 'CD63', 'MFGE8', 'GLUL', 'PPDPF', 'SREBF1', 'MAGED2', 'DEGS2', 'C12orf57', 'RPSA', 'SPINT2', 'RPL41', 'RHOC'', 'SLC44A2', 'SERINC', 'PTMS', 'COX6B1', 'CTSB', 'UBB' 'WDR83OS', 'HNRNPA2B1', 'ATPSFIE', 'CIQB', 'GRN', 'COL6A2', 'SEC61A1', 'ATG10', 'EIFSA', 'CRABP2', 'ELOB', 'TI', 'COX4I1', 'SELENOW', 'TXNIP", 'UQCRI', 'RPL6', 'MMP14', 'MDK', 'GUKI', 'TUBB', 'H1-2', 'LMNA', 'RPL22', 'MYL6, 'CHCHD2', 'CTTN', 'RPL10A', 'EEFIAI', 'LSM4', 'DHCR24', 'CAPNSI', 'LAPTM4A', 'SSR2', 'UBE2M', 'TSPO', 'LMAN2', 'RPS10', 'PFDNS", 'NENF, 'TSKU', 'SERPINHI', 'IGLLI', 'LGALSI', 'CHPF, 'EEFID', 'MIDN', 'ATPSMC2', 'CLDN7', 'STARD3', 'LLGL2', 'CCT3'" 'EIF3B', 'ZNF90', 'SH3BGRL3', 'MZBI', 'CDC37', 'PTBP1', 'AP2MI', 'IGFBP7', 'ENSA', 'EIF4G1' |

Figure 6: **Selected high variance genes for gene expression prediction task across datasets.**

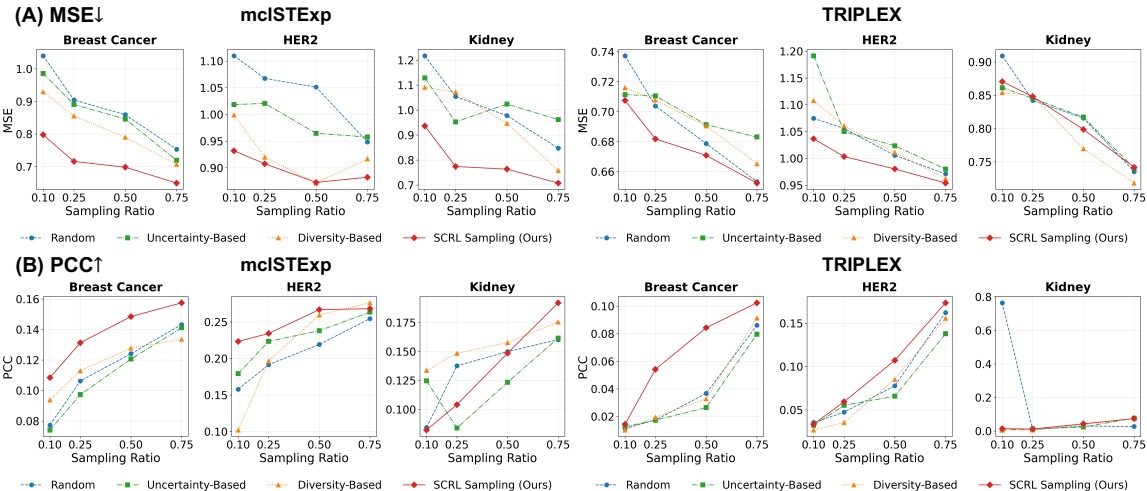

Figure 7: **Comparison of sampling strategies across multiple methods on different datasets.** We evaluate performance under training data ratios of 10%–75% to simulate low-resource scenarios, with each curve denoting a sampling strategy. Our proposed SCRL sampling consistently achieves better performance across datasets and models, particularly in low-budget (10%–25%) regimes.

## Appendix B. Additional Implementation Details

To evaluate the effectiveness of our sampling strategy, we compare it against three representative baselines: Monte Carlo random sampling, uncertainty-based sampling (Safaei et al., 2024), and diversity-driven sampling (Zhdanov, 2019).

**Uncertainty-based sampling.** This method estimates prediction uncertainty via Monte Carlo Dropout. Specifically, we insert a Dropout layer (drop rate = 0.1) after the feature extraction block of the vision encoder, and the model outputs the expression values of 300 genes. During uncertainty estimation, the Dropout layer remains activated, and we perform $T = 20$ stochastic forward passes for each patch. We compute the variance across these predictions and use its mean as the entropy score. Higher entropy indicates greater model uncertainty, and patches with high entropy are prioritized during sampling.

Table 4: **Summary of computational cost.** We summarize the parameter size, running time per epoch, GPU memory consumption, and number of training samples of our model on Breast Cancer dataset.

| Module | Parameter | Time/epoch | GPU Mem | Samples |
|---|---|---|---|---|
| SCR$^2$Net | 33.82 M | 85.70 s | 35.13 GB | 27,171 |
| SCRL | 0.6 M | 50.54 s | – | 3,099,206 |

**Diversity-driven sampling.** This method encourages sample diversity based on feature similarity. We first extract 1024-dimensional visual features using the vision encoder during

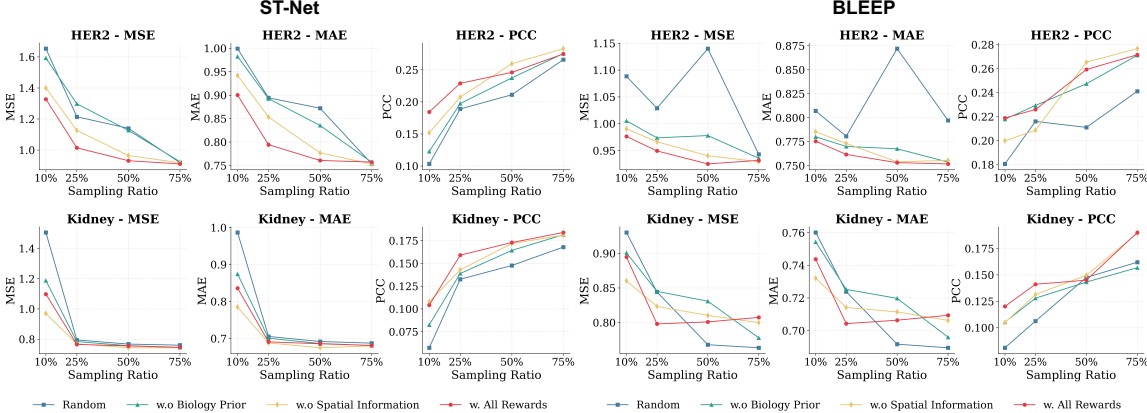

Figure 8: **Ablation study of reward function in our proposed SCRL sampling.**

training. These features are standardized and reduced to 128 dimensions via PCA, followed by clustering using DBSCAN. To avoid insufficient cluster granularity, we incorporate a dynamic adjustment mechanism: the minimum cluster count is adaptively set to $\sqrt{N}/5$, where $N$ denotes the total number of samples. If DBSCAN yields too few clusters, we automatically switch to KMeans to enforce the desired number of clusters. During sampling, patches are drawn uniformly from each cluster to ensure diverse coverage of tissue regions.

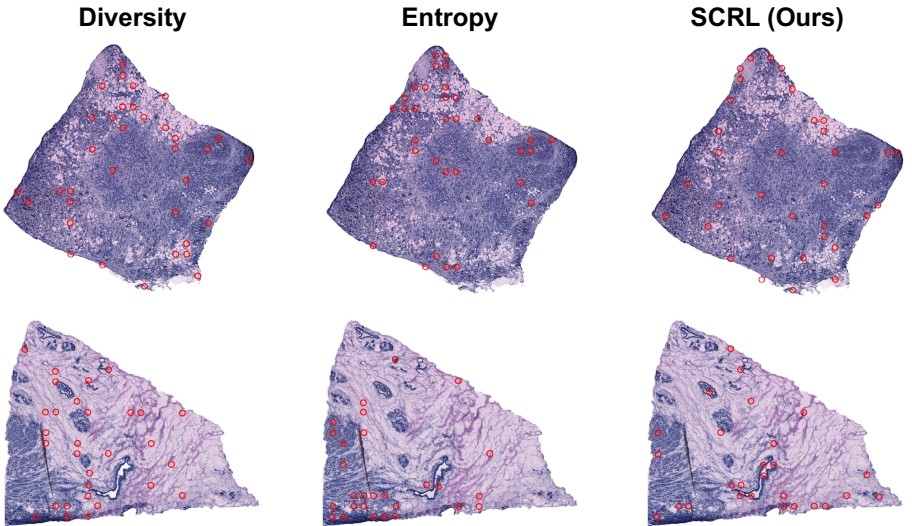

Figure 9: **Spatial distribution of sampled spots under 10% sampling.** Red circles indicate selected spots on whole-slide images. Entropy-based sampling produces concentrated regions, while diversity-based sampling wastes samples in low-density background areas. Our method (SCRL) achieves more informative and balanced sampling by incorporating spatial and biological cues.

**Our SCRL sampling strategy.** For the multi-objective reward function, we set $w_{\text{sc}} = 20$, $w_{\text{type}} = 5$, and $w_{\text{spa}} = 0.05$ to balance manifold coverage, cell-type diversity, and spatial density constraints. The loss weights $\lambda_r$, $\lambda_p$, and $\lambda_{KD}$ are set to 1.0, 0.25, and 0.25, respectively. We adopt a similarity confidence mask with threshold $m = 0.15$. Active sampling proceeds for 20 rounds, with the sampled training set updated every 5 epochs. The initial round employs random sampling for warm-up. A fixed seed of 42 is used for reproducibility. During retrieval, we select the top-50 most similar expression profiles and retain only the top-10 dominant cell types for robust reference aggregation.

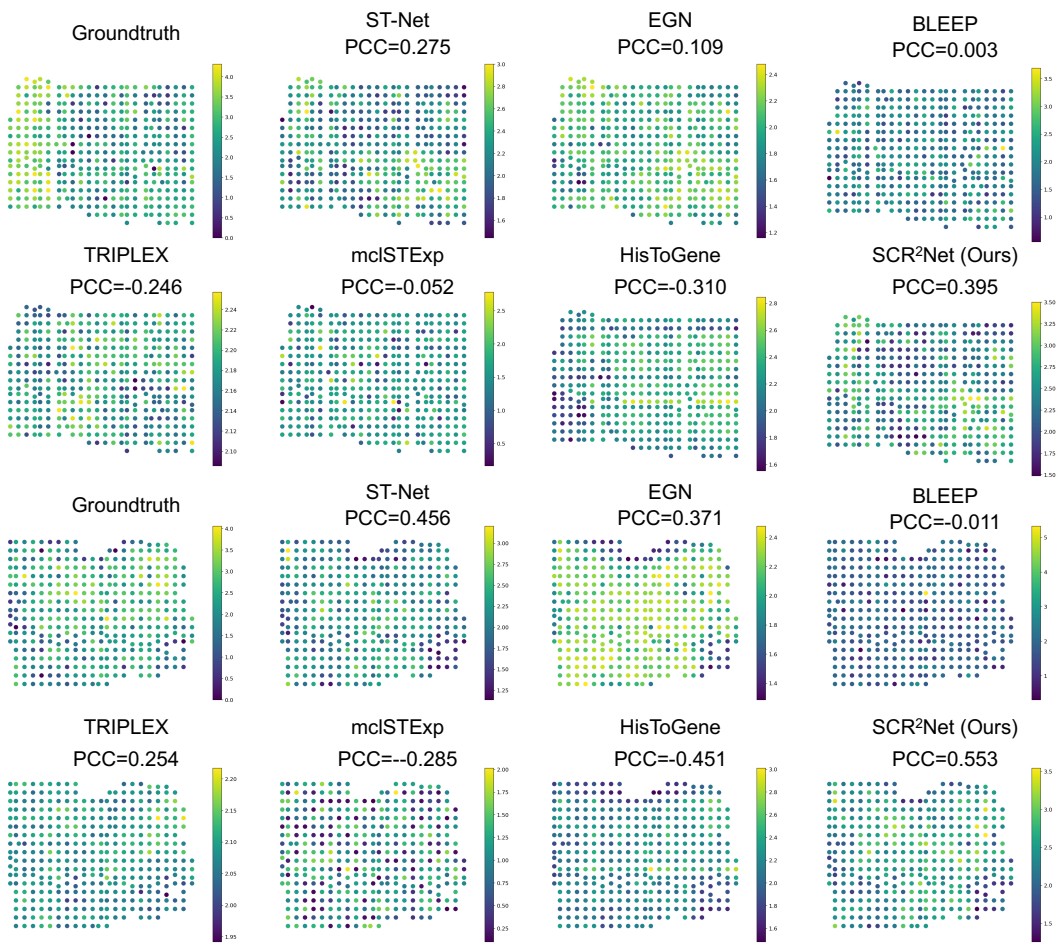

Figure 10: Visualization of predicted spatial expression distribution of cancer-related gene RPS3 on WSIs.

## Appendix C. Additional Experimental Results

Due to space limitations in the main manuscript, we provide additional experimental results in this appendix, including the qualitative analysis of low-budget sampling (Figure 9),

the comparison of sampling strategies evaluated on mlSTExp and TRIPLEX benchmarks (Figure 7), and an ablation study on the reward function design (Figure 8).

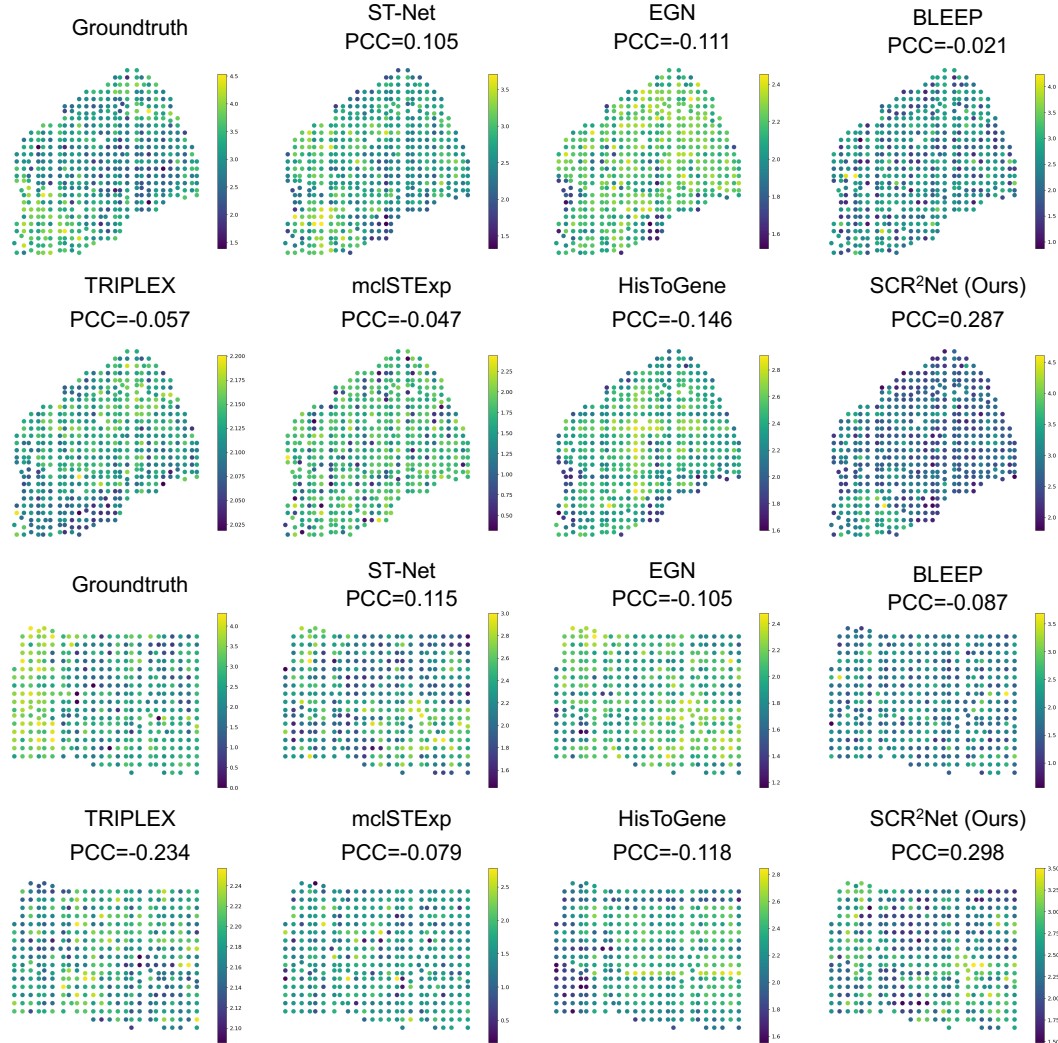

Figure 11: Visualization of predicted spatial expression distribution of cancer-related gene RPL19 on WSIs.

**Qualitative analysis of low-budget sampling.** Figure 8 further provides a qualitative comparison of different sampling strategies by visualizing the spatial distribution of sampled spots on whole-slide images under a low sampling budget (10%). As shown in the figure, entropy-based sampling tends to select spots from relatively concentrated regions, resulting in limited spatial coverage of the tissue. In contrast, diversity-based sampling distributes samples more broadly, but often allocates a substantial portion of the limited sampling budget to low tissue-density or background regions, leading to inefficient sampling. By

jointly incorporating spatial relationships and biological information, our method achieves a more balanced sampling pattern that focuses on informative tissue regions while maintaining adequate spatial coverage. This behavior explains the superior performance of our approach under low-budget sampling settings, where effective utilization of limited sampling resources is critical.

**Cancer-related Gene Expression Prediction.** We further selected two genes that are highly relevant to breast cancer, RPL19 and RPS3. RPL19 is frequently overexpressed in breast cancer tissues, making it a commonly used reference and prognostic-related gene in breast cancer studies (Hong et al., 2014). RPS3's dysregulation is associated with enhanced tumor aggressiveness and poor prognosis in breast cancer (Ono et al., 2017). We visualize the predicted spatial expression patterns of these two genes on whole-slide images in Figure 11 and Figure 10, demonstrating that the proposed model captures spatially coherent and biologically meaningful expression distributions, thereby highlighting its interpretability at clinically relevant molecular levels.

## Appendix D. Practical Considerations and Future Directions

In realistic scenarios, the proposed SCRL sampler can start from any amount of existing spatial transcriptomics (ST) data and does not require sampling tissues at full coverage. The sampling process is formulated as a sequential decision procedure, in which the sampler is optimized online based on the information observed so far, and adaptively selects tissue locations to query until the predefined sampling budget is reached.

A pretrained sampler can be directly transferred to unseen data from a new laboratory for sampling, and does not require collecting any amount of ST data as a start point on the new dataset. This allows the sampler to be applied without introducing additional experimental overhead at the initial stage. As more data become available in future studies, the sampler can be further optimized, which is expected to improve its robustness and transferability across different experimental settings.

