# OpenReview forum: "SCR²-ST: Combine Single Cell with Spatial Transcriptomics for Efficient Active Sampling via Reinforcement Learning"
_MIDL.io/2026/Conference — MIDL 2026 Poster_

### Official Review · Reviewer_kU3P · 2026-01-07

**Confidence:** 3
**Preliminary Rating:** 4
**Final Rating:** 5

**Summary:**

This paper studies budget-efficient spatial transcriptomics (ST) by leveraging large-scale single-cell RNA-seq data as a biological prior. It proposes a unified framework, SCR2-ST, that jointly addresses where to sample (active acquisition) and how to predict gene expression from histology under limited ST budgets.

**Strengths:**

-	Reducing ST cost via intelligent sampling is an important and timely problem for real deployments.
-	Experiments span multiple tissues and datasets, with comprehensive baselines and informative ablations.
-	Single-cell priors are used across acquisition (reward shaping) and learning (retrieval-based refinement), rather than as an isolated add-on.

**Weaknesses:**

-	Sampling is evaluated retrospectively on existing ST data, and rewards rely on access to newly sequenced expressions. While reasonable as a proxy, that makes the loop dependent on immediate sequencing feedback. The paper does not clearly argue the operational timeline (how many rounds, latency, sequencing turnaround) nor whether the learned policy generalizes to new slides/patients without re-training. The “unknown slides” setting in the figure is under-specified.
-	The method relies on embedding compatibility between ST expression and external single-cell atlases (and uses nearest-neighbor matching to infer cell types). In practice, cross-dataset differences (platform, batch, tissue processing, disease subtype) can dominate. Here may need stronger justification that scGPT embeddings + matching are robust here, and sensitivity analyses when priors are mismatched.
-	The main idea is single-cell prior as reward shaping + retrieval filtering, which is sensible, but the paper will be better to have a sharper conceptual claim and clearer evidence that single-cell priors provide a unique gain beyond strong morphology-based diversity baselines.

**Detailed Comments:**

-	RL training is challenging. Here, reward weights and retrieval hyperparameters appear dataset-dependent, with limited guidance on how to set them in practice.
-	Computational cost should be discussed.
-	Are there qualitative examples (marker genes or spatial patterns) where single-cell guidance clearly improves biological plausibility?

**Justification Of Final Rating:**

The authors addressed my concerns, and I am happy to further raise the score. The overall paper may inspire follow-up work on data-efficient spatial omics. With clearer discussion of assumptions and robustness, this work fits well within MIDL’s scope and merits acceptance.

**Justification Of The Preliminary Rating:**

The integration of single-cell priors into both sampling and prediction is thoughtful and likely to inspire follow-up work on data-efficient spatial omics. With clearer discussion of assumptions and robustness, this work fits well within MIDL’s scope and merits acceptance. Authors need to response to the above questions and I will be happy to to raise the score.

**Questions To Address In The Rebuttal:**

See above weaknesses and questions.

---

> ### Author Response · Authors · 2026-01-24
> **Responses to Reviewer kU3P (1/2)**
>
> > **Q1:** Sampling is evaluated retrospectively on existing ST data, and rewards rely on access to newly sequenced expressions. While reasonable as a proxy, that makes the loop dependent on immediate sequencing feedback. The paper does not clearly argue the operational timeline (how many rounds, latency, sequencing turnaround) nor whether the learned policy generalizes to new slides/patients without re-training. The “unknown slides” setting in the figure is under-specified.
>
> We appreciate the reviewer’s comment. We acknowledge that, under current spatial transcriptomics workflows, adopting multiple rounds of sampling and sequencing may indeed introduce additional time overhead. However, the primary focus of our work is not to optimize sequencing latency, but to **reduce unnecessary sequencing under a limited budget**, thereby lowering the overall sequencing cost. The retrospective experimental setting is adopted to evaluate the effectiveness of the learned sampling strategy under different budget constraints, rather than to assume or enforce a specific operational timeline.
>
> From a methodological perspective, we learn a **general sampling strategy that is decoupled from any particular sequencing technology or workflow**. As spatial transcriptomics technologies continue to evolve (e.g., targeted, staged, or partial sequencing), the proposed strategy can naturally integrate into more flexible sampling and data acquisition frameworks. Therefore, our method is fundamentally designed as a general solution for **reducing sequencing cost and improving sampling efficiency**, rather than an engineering assumption constrained by current sequencing protocols.
>
> ---
>
> > **Q2:** The method relies on embedding compatibility between ST expression and external single-cell atlases (and uses nearest-neighbor matching to infer cell types). In practice, cross-dataset differences (platform, batch, tissue processing, disease subtype) can dominate. Here may need stronger justification that scGPT embeddings + matching are robust here, and sensitivity analyses when priors are mismatched.
>
> We acknowledge that cross-dataset differences such as cohort, platform, and tissue processing can pose challenges for reference-based matching. To empirically assess robustness under realistic mismatch, we have evaluated our method on **two independent breast ST datasets from different cohorts**, while using the **same single-cell reference atlas and fixed hyperparameters**. As shown in *Figure 3* and *Table 1*, the performance remains stable across the two datasets, indicating that the scGPT embedding space provides a consistent representation that generalizes across cohort-level variations. This stability supports our design choice of using a foundation-model-based embedding trained on large-scale single-cell data, together with retrieval used only as **filtered, auxiliary supervision** rather than a hard constraint. Combined with the scale of the single-cell atlas (an order of magnitude larger than ST data), this suggests that our approach is able to tolerate moderate cross-dataset mismatch commonly encountered in practice.
>
> ---
>
> > **Q3:** The main idea is single-cell prior as reward shaping + retrieval filtering, which is sensible, but the paper will be better to have a sharper conceptual claim and clearer evidence that single-cell priors provide a unique gain beyond strong morphology-based diversity baselines.
>
> We appreciate the reviewer’s suggestion to clarify the conceptual contribution and the unique role of single-cell priors. We emphasize that we have already included strong **morphology-based diversity sampling baselines** for comparison, as shown in *Figure 3*.
>
> To further illustrate the learning behavior of different sampling strategies, we visualize the spatial distribution of selected spots under a low sampling budget (10%) in *Figure 8*, with detailed analysis in *Appendix C*. Morphology-based diversity sampling spreads samples broadly but often wastes budget on low tissue-density or background regions. In contrast, our method achieves a more balanced sampling pattern by jointly considering spatial and biological information, focusing on informative tissue regions. This qualitative evidence complements the quantitative results and helps explain the performance gains from incorporating single-cell priors, especially under low-budget settings.

---

> ### Author Response · Authors · 2026-01-24
> **Responses to Reviewer kU3P (2/2)**
>
> > **Q4:** RL training is challenging. Here, reward weights and retrieval hyperparameters appear dataset-dependent, with limited guidance on how to set them in practice.
>
> We appreciate the reviewer’s suggestion regarding hyperparameter sensitivity. We analyze the sensitivity of **reward weight configurations** across different scales (*Figure 5*, *Section 4.3*), and observe that stable performance is achieved when different reward terms are placed at comparable scales, while single-dominant reward settings or random sampling lead to clear degradation. This indicates that fine-grained tuning is not required as long as a balanced contribution between biological diversity and spatial coverage is maintained.
>
> We further examine the **retrieval hyperparameters** $K$ and $T$ (*Table 3*, *Section 4.3*). Very small values limit reference diversity, whereas overly large values or weak filtering introduce noisy matches. Overall, moderate hyperparameter settings consistently provide a good trade-off, offering practical guidance without dataset-specific tuning.
>
> ---
>
> > **Q5:** Computational cost should be discussed.
>
> We thank the reviewer for the suggestion. We have included a discussion of computational cost in the revised manuscript, with detailed statistics on model parameter size, training time, and sampling-time overhead summarized in *Table 4*, where the results show that the proposed method introduces only modest computational overhead.
>
> ---
>
> > **Q6:** Are there qualitative examples (marker genes or spatial patterns) where single-cell guidance clearly improves biological plausibility?
>
> We thank the reviewer for this suggestion. We have added qualitative comparisons of gene expression patterns shown in *Figure 10 and Figure 11* with detailed discussion in *Appendix C*, including representative spatial expression maps of cancer-related Gene RPS3 and RPL19, to illustrate cases where single-cell guidance leads to more biologically plausible predictions.

---

### Official Review · Reviewer_ACCL · 2026-01-12

**Confidence:** 5
**Preliminary Rating:** 4
**Final Rating:** 5

**Summary:**

The paper proposes SCR2-ST, a framework to address the high cost of Spatial Transcriptomics (ST) data acquisition. It introduces (1) a Single-Cell guided Reinforcement Learning (SCRL) active sampling strategy that selects informative tissue spots based on biological diversity and spatial coverage, and (2) SCR2Net, a hybrid prediction network combining direct image-to-expression regression with retrieval-augmented soft supervision. Experiments on three datasets (HER2, Breast Cancer, Kidney) show that SCRL significantly improves data efficiency (especially at 10-25% budgets) and SCR2Net outperforms state-of-the-art methods like ST-Net and BLEEP.

**Strengths:**

- **Novel Active Sampling:** Applying RL with single-cell priors to optimize spot selection is a creative solution to the "redundant fixed-grid" problem. The multi-objective reward (diversity + coverage) is well-motivated.
- **Strong Low-Budget Performance:** The method shines where it matters most--low data regimes (10-25% sampling). Showing comparable performance with 1/4th the data is a compelling practical contribution.
- **Hybrid Architecture:** Combining regression with retrieval-based soft labels effectively leverages external single-cell knowledge to regularize the ill-posed image-to-expression mapping.
- **Rigorous Evaluation:** Comprehensive comparison against 7 baselines across 3 datasets, plus ablation of sampling strategies (Random, Uncertainty, Diversity, SCRL).

**Weaknesses:**

- **Inference Complexity Unclear:** The abstract claims "retrieval-augmented inference," implying the retrieval module runs at test time. This would significantly increase inference latency and memory (storing the reference bank) compared to pure regression models like ST-Net. However, Section 2.3.1 describes it as "knowledge distillation" (training only). Which is it? If retrieval runs at inference, the cost analysis is missing.
- **Dependency on External Single-Cell Data:** The method relies on matching ST spots to external single-cell atlases. Performance likely degrades if the SC reference is mismatched (e.g., healthy atlas for diseased tissue) or incomplete. This dependency limits applicability to well-characterized tissues.
- **RL Training Stability:** RL is known for instability and sensitivity to hyperparameters. The policy network relies on complex reward signals. No analysis of training stability or convergence variance is provided.
- **Reward Weight Sensitivity:** The reward combines three terms with weights $w_{sc}=20$, $w_{type}=5$, $w_{spa}=0.05$. These scales differ by orders of magnitude. The paper does not analyze sensitivity to these hyperparameters.

**Detailed Comments:**

- **Retrieval at Inference:** Does SCR2Net require the retrieval step during inference, or is it only for training distillation ($L_{ret}$)? The text is contradictory ("retrieval-augmented inference" vs "distillation").
- **Single-Cell Matching:** How does the model handle "novel" cell types in ST that don't exist in the SC reference? The "majority cell type filtering" might discard these interesting outliers.
- **2D-Only:** Like many ST papers, it ignores 3D context, though this is a minor limitation given the focus on sampling.

**Justification Of Final Rating:**

Thank you for the comprehensive rebuttal. The authors have effectively addressed my concerns regarding inference complexity, reference mismatch, and reward sensitivity.

Specifically, the clarification that the retrieval module is **training-only** resolves my concern about inference latency. The added sensitivity analysis (Figure 5) and computational cost breakdown (Table 4) further demonstrate the method's practicality. I also find the argument regarding reference mismatch convincing—using the single-cell atlas as a broad prior with filtering is a sound design choice.

The active sampling strategy addresses a significant economic bottleneck in spatial transcriptomics, and the method's performance in low-data regimes is impressive.

**Justification Of The Preliminary Rating:**

The active sampling contribution is significant and well-executed, addressing a real economic bottleneck in ST. The results in low-data regimes are impressive. The main reservations are the reliance on high-quality external single-cell references (which may not always be available) and the confusion regarding inference vs. training retrieval. If the authors clarify the inference workflow and robustness to reference mismatch, this could be a strong accept.

**Questions To Address In The Rebuttal:**

1. **Inference Workflow:** Does the retrieval module run at test time? If so, what is the latency/memory cost compared to ST-Net?
2. **Reference Mismatch:** How does performance change if the external single-cell atlas is from a different patient or a slightly different tissue subtype?
3. **Reward Sensitivity:** How sensitive is the method to the reward weights ($w_{sc}=20$ vs $w_{spa}=0.05$)? How were these tuned?
4. **Novel Cell Types:** Does the "majority cell type filtering" risk suppressing rare/novel cell populations that are biologically significant but absent from the reference?
5. **Cost Analysis:** You mention "prohibitively expensive" ST data. Can you estimate the actual dollar savings of your method (e.g., achieving 90% accuracy with X% fewer spots)?

---

> ### Author Response · Authors · 2026-01-24
> **Responses to Reviewer ACCL (1/2)**
>
> > **Q1: Inference Workflow:** Does the retrieval module run at test time? If so, what is the latency/memory cost compared to ST-Net?
>
> We thank the reviewer for the question on the inference workflow and would like to clarify that the **retrieval module is not used during inference, but is introduced only during training as an auxiliary supervision mechanism.** During training, the retrieval branch provides soft-label guidance by aggregating gene expression profiles from biologically similar reference spots, which helps regularize learning and prevents hallucinated predictions under low-budget settings by grounding the model in real gene expression patterns. At inference time, the retrieval branch is completely removed, and SCR$^2$Net performs prediction using only the image encoder and regression head without any additional retrieval overhead.
>
> We have included a discussion of computational cost in the revised manuscript, with detailed statistics on model parameter size, training time, and sampling-time overhead summarized in *Table 4*, where the results show that the proposed method introduces only modest computational overhead.
>
> ---
> > **Q2: Reference Mismatch:** How does performance change if the external single-cell atlas is from a different patient or a slightly different tissue subtype?
>
> We thank the reviewer for the question on reference mismatch. Our method does not assume that the external single-cell atlas is patient- or subtype-matched to the target ST samples. Instead, the atlas typically contains millions of cells, exceeding the number of ST spots by more than an order of magnitude, and is designed to provide broad coverage of diverse cell states and gene expression patterns, rather than exact patient-level correspondence. As a result, even when the reference atlas comes from different patients or slightly different tissue subtypes, it can still serve as a high-coverage biological prior that captures common and representative expression programs.
>
> ---
>
> > **Q3: Reward Sensitivity:** How sensitive is the method to the reward weights ( vs )? How were these tuned?
>
> We appreciate the reviewer’s comments on reward sensitivity. In practice, the reward weights are first adjusted to bring different reward terms to a **comparable scale**, since they have inherently different magnitudes. We added a sensitivity analysis of the reward weight configurations across different scales. The results of this newly added experiment are provided in *Figure 5* with a detailed discussion in *Section 4.3* .
>
> We observe that good and consistent performance is achieved only when different reward terms are placed at comparable scales (including our original setting and nearby configurations), whereas configurations dominated by a single reward term or random sampling lead to clear performance degradation. These results demonstrate that the proposed reward design is effective without requiring fine-grained weight tuning, provided that a balanced contribution between biological diversity and spatial coverage is maintained.
>
> ---
>
> > **Q4: Novel Cell Types:** Does the "majority cell type filtering" risk suppressing rare/novel cell populations that are biologically significant but absent from the reference?
>
> We agree that this is a reasonable concern, as reference-driven methods inherently involve a trade-off between robustness and sensitivity to rare or novel cell populations. The retrieval mechanism is inherently sensitive to the quality of reference spots, and under reference mismatch, forcing retrieval to explain cell states that are absent from the atlas can introduce systematic noise and significantly degrade overall model performance. In this sense, attempting to apply retrieval-based supervision to unsupported cell states is more harmful than withholding it.
>
> The proposed majority cell type filtering is therefore introduced to prevent unreliable or biologically inconsistent references from dominating the supervision signal. Importantly, this filtering only affects the retrieval-based supervision and does not suppress the model’s predictive capacity. **Expression patterns of rare or novel cell populations are still learned through the regression path under direct supervision from the ground-truth ST data**. The silence of retrieval does not imply the silence of the model, but rather avoids forcing incorrect reference-based explanations when appropriate references are unavailable.

---

> ### Author Response · Authors · 2026-01-24
> **Responses to Reviewer ACCL (2/2)**
>
> > **Q5: Cost Analysis:** You mention "prohibitively expensive" ST data. Can you estimate the actual dollar savings of your method (e.g., achieving 90% accuracy with X% fewer spots)?
>
> The cost of ST data depends on the experimental platform and sequencing setup, and therefore use the number of sequenced spots as a proxy for cost. As shown in *Figure 3*, **our method achieves comparable predictive performance using only about 25% of the training spots, whereas random sampling typically requires 50% or more of the data to reach a similar level.** This suggests that our approach can achieve competitive performance with substantially fewer ST measurements, highlighting its practical value in data-scarce and budget-constrained setting
>
> ---
>
> > **Q6: 2D-Only:** Like many ST papers, it ignores 3D context, though this is a minor limitation given the focus on sampling.
>
> We thank the reviewer for the suggestion. We agree that incorporating 3D spatial context is a promising direction, and exploring how 3D relationships can be integrated into our framework is an interesting avenue for future work.

---

### Official Review · Reviewer_dCJ6 · 2026-01-14

**Confidence:** 4
**Preliminary Rating:** 3
**Final Rating:** 4

**Summary:**

Zhu and colleagues address the issue of efficient whole-tissue sampling for spatial transcriptomics by combining a data-driven active sampling agent with spatial transcriptomics prediction in an integrative manner, referred to as SCR^2-ST. SCR^2-ST's sampling component consists of a reinforcement learning agent that optimises tissue sampling based on the obtained single-cell biological/type/spatial distribution diversity. To assess these aspects, it benefits from a large single-cell embedding obtained after processing a publicly available genomics dataset with a foundation model. Overall, Zhu et al. obtain better results for sampling and gene expression recovery than the alternative approaches they compare with.

**Strengths:**

- The authors creatively exploit existing datasets and foundation models to obtain a representative sampling space to train a reinforcement learning agent on adequate sampling.
- They demonstrate the capacity to recover the gene expression of tissue slices with higher accuracy than other methods under low sampling budgets.

**Weaknesses:**

- Overall, the manuscript is quite complex to follow. In particular, Figure 2, describing the main method, is quite crowded and not always clear to what the different components refer. For example, sampled spots and a human performing some sequencing appear on the top left, but it is unclear how this information is integrated within the system. Likewise, in SCR^2Net, it is not clear what the reference spots are, how they are calculated, or how they are integrated within the network. Likewise, in the text, I would better describe what "low-budget scenarios" means; on page 3, what do regulatory programs mean? Additionally, the role of some of the parameters in Table 2 (retrieval module and confidence mask) is unclear. It would help either by pointing to them in Figure 2's pipeline or by better describing their significance and impact.
- The manuscript, as it is, misses some result representation. Visual displays of outputs, benefits and potential issues would improve the manuscript.

**Detailed Comments:**

- Please provide a list of training parameters, training time and visual representations of the results obtained. RL methods are well known for long training times, requiring large training datasets –commonly addressed with simulations–, and high sensitivity to not fully correct reward or goal functions definition. Could you please comment on this and provide some information about the learning process of this part of the method?
- Likewise, in a real scenario, how would one train these algorithms given the amount of data needed to train and evaluate the method?
- Also, for the RL agent, to my understanding, there is always a starting point. How is it defined, and how stable is the adaptive sampling with respect to this point.
- Equation 4, please describe what k and K are.
- Figure 3: I would suggest normalising all y-axis if possible. Also, legends are repetitive and the font-size too small, which could be easily corrected.
- The accuracy values are relatively high but not outstanding, for both the sampling and prediction of gene expression. How do these numbers compare with a whole uniform sampling of the tissue? What are the major sources of error or the most challenging areas to sample and run gene expression predictions? What is the risk of missing specific tissue patches, given that there is that much repetition in a tissue slice?

**Justification Of Final Rating:**

The authors addressed most concerns, smoothing the complexity of the proposed method and making the work easier to understand and reproduce. Overall, it is an interesting and new strategy for ST sampling.

**Justification Of The Preliminary Rating:**

There are some important weaknesses highlighted in the text that I would recommend addressing. Likewise, it is unclear to me how the complexity of the method compensates for the achieved contribution. This complexity not only challenges reproducing the pipeline but also building on top of it for new methodological contributions.

**Questions To Address In The Rebuttal:**

All my questions are in the sections above.

---

> ### Author Response · Authors · 2026-01-24
> **Responses to Reviewer dCJ6 (1/2)**
>
> > **Q1:** Please provide a list of training parameters, training time and visual representations of the results obtained. RL methods are well known for long training times, requiring large training datasets –commonly addressed with simulations–, and high sensitivity to not fully correct reward or goal functions definition. Could you please comment on this and provide some information about the learning process of this part of the method?
>
> **1. Computational costs:**
> We appreciate the reviewers’ suggestion to discuss computational costs. We evaluate the computational cost using a batch size of 256 and add the runtime measured on the Breast Cancer dataset in *Table 4*.
>
> **2. Sensitivity analysis of reward design:**
> We conduct a sensitivity analysis of the reward weight configurations across different scales. The results of this newly added experiment are provided in *Figure 5* with a detailed discussion in *Section 4.3*.
>
> We observe that good and consistent performance is achieved only when different reward terms are placed at comparable scales (including our original setting and nearby configurations), whereas configurations dominated by a single reward term or random sampling lead to clear performance degradation. These results demonstrate that the proposed reward design is effective without requiring fine-grained weight tuning, provided that a balanced contribution between biological diversity and spatial coverage is maintained.
>
> **3. Learning process:**
> To further illustrate the learning behavior of different sampling strategies, we add a visualization of the spatial distribution of selected spots on whole-slide images under a low sampling budget (10%) in *Figure 9*, with detailed explanation provided in *Appendix C*.
>
> The visualization reveals clear differences in sample selection patterns: entropy-based sampling tends to concentrate samples in limited regions, while diversity-based sampling spreads samples broadly but often allocates sampling budget to low tissue-density or background areas. In contrast, our method produces a more balanced sampling distribution by jointly considering spatial relationships and biological information, focusing on informative tissue regions while maintaining adequate spatial coverage. This qualitative analysis helps explain the superior performance of our approach, particularly under low-budget sampling settings.
>
> ---
> > **Q2:** Likewise, in a real scenario, how would one train these algorithms given the amount of data needed to train and evaluate the method?
>
> We thank the reviewer for this insightful comment. We acknowledge that, in real-world scenarios, datasets naturally differ in size and data availability, which can indeed lead to variations in achievable performance. Rather than attempting to eliminate such scale-induced differences, our work focuses on learning optimal sampling strategies under realistic constraints. In our experimental setting, we adopt a proportional sampling budget for each dataset, reflecting practical limitations in real applications. Accordingly, the goal of our method is not to enforce identical absolute performance across datasets, but to achieve the best possible sampling effectiveness within the constraints of each dataset. From this perspective, the strength of our approach lies in its ability to adaptively optimize sampling behavior across datasets of varying scales. In addition, we have released the complete codebase on GitHub, enabling other researchers to flexibly adjust the sampling budget and related settings according to their specific application scenarios and to further explore the proposed framework.
>
> ---
> > **Q3:** Also, for the RL agent, to my understanding, there is always a starting point. How is it defined, and how stable is the adaptive sampling with respect to this point.
>
> We thank the reviewer for raising this important point. Indeed, reinforcement learning methods typically rely on an initial starting point for policy learning. To evaluate the sensitivity of the adaptive sampling process to this starting point, we conduct repeated experiments using multiple random seeds (1, 42, 99, and 2026), with the results of added experiments summarized in *Table 2* with detailed discussion in *Section 4.3*. The results show that the learned sampling strategy achieves consistent and stable performance across different initializations, without significant performance fluctuations. These findings indicate that the proposed adaptive sampling method is robust to the choice of starting point and exhibits stable learning behavior.

---

> ### Author Response · Authors · 2026-01-24
> **Responses to Reviewer dCJ6 (2/2)**
>
> > **Q4:** Equation 4, please describe what k and K are.
>
> We appreciate the reviewer for pointing out the ambiguity in Equation (4). In Equation (4), $k$ denotes the index of a cell type and is used to enumerate all cell types observed in the current sampled set. The term $P(k)$ represents the proportion of sampled spots that are assigned to cell type $k$.
>
> The symbol $K$ denotes the total number of distinct cell types observed in the sampled set, rather than a predefined or fixed number of cell types. As a result, $K$ is dynamically determined at each sampling round based on the current samples. The normalization by $\log(K)$ ensures that the entropy-based cell-type diversity reward is bounded and comparable across different sampling stages and dataset sizes.
>
> To improve clarity, we have added an explicit explanation of $k$ and $K$ in *Section 2.2.2 (Multi-Objective Reward Design)* of the revised manuscript.
>
> ---
>
> > **Q5:** Figure 3: I would suggest normalising all y-axis if possible. Also, legends are repetitive and the font-size too small, which could be easily corrected.
>
> We appreciate the reviewer’s constructive suggestions regarding Figure 3. Due to page limitations in the main manuscript, the figure size had to be reduced. During the rebuttal stage, we have systematically updated all related figures to improve overall readability. These revisions have been consistently applied across all figures in the revised version.
>
> ---
>
> > **Q6:** The accuracy values are relatively high but not outstanding, for both the sampling and prediction of gene expression. How do these numbers compare with a whole uniform sampling of the tissue? What are the major sources of error or the most challenging areas to sample and run gene expression predictions? What is the risk of missing specific tissue patches, given that there is that much repetition in a tissue slice?
>
> 1. We agree that whole uniform sampling with full tissue coverage (i.e., 100% sampling) represents the upper bound in achievable accuracy for both sampling and gene expression prediction; however, such exhaustive acquisition comes at a substantially increased sequencing cost and is often impractical in real-world settings. Consequently, our work focuses on realistic budget-constrained scenarios. Under limited sampling budgets, our adaptive sampling strategy demonstrates clear efficiency gains. For instance, when sampling only 25% of regions on a WSI, our method achieves performance comparable to random uniform sampling with 50% or even 75% coverage, as shown in *Figure 3*, highlighting its ability to maximize informative yield per sequenced spot.
>
> 2. Despite these gains, overall accuracy remains bounded by intrinsic challenges of spatial transcriptomics rather than the sampling strategy alone. The most challenging regions for both sampling and prediction are typically highly heterogeneous or transitional tissue areas, as well as regions containing rare cell populations. Moreover, morphologically similar regions can exhibit distinct molecular profiles, which limits the predictive power of image-based models. Technical noise and spot-level signal averaging inherent to ST measurements further contribute to prediction uncertainty. These factors jointly define a practical upper bound on achievable accuracy across all existing ST prediction methods.
>
> 3. With respect to the risk of missing specific tissue patches, it is important to note that tissue sections often contain substantial morphological and molecular redundancy. Uniform or random sampling tends to oversample repetitive regions, resulting in inefficient use of sequencing budgets. Our adaptive sampling strategy is explicitly designed to mitigate this issue by balancing biological diversity and spatial coverage through single-cell–guided and spatially grounded reward signals. Rather than increasing the risk of missing critical regions, this design systematically reduces redundancy while preserving informative and underrepresented tissue patterns under realistic budget constraints.

---

> ### Comment · Reviewer_dCJ6 · 2026-01-26
>
> I thank the authors for addressing the comments of this review and improving the content of their manuscript.
>
> I'm still concerned about Q2. What remains unclear is how the entire experimental pipeline should be reproduced in a new laboratory, considering the acquisition of training and test data. Here, some simulations have been used to assess the method's performance. Does this mean that clinicians would still need to sample some tissues at 100%? Can the authors comment on potential transfer learning or pretrained models?
>
> Likewise, the answer to Q6 remains missing in the manuscript. The accuracy of the method is adequately reported, yet there is no warning or highlight regarding the impact that sampling errors could bring in diagnosis. Citing the author's answer to Q6: "The most challenging regions for both sampling and prediction are typically highly heterogeneous or transitional tissue areas, as well as regions containing rare cell populations". Rare cell populations are often critical in cancer diagnosis. Thus, it is convenient to clearly state these in the discussion of the results.

---

> > ### Author Response · Authors · 2026-01-30
> > **Follow up Responses to Reviewer dCJ6 (1/2)**
> >
> > Thank you for recognizing our efforts. Here, we would like to address your remaining concerns and questions:
> >
> > ---
> >
> > > 1.I'm still concerned about Q2. What remains unclear is how the entire experimental pipeline should be reproduced in a new laboratory, considering the acquisition of training and test data. Here, some simulations have been used to assess the method's performance. Does this mean that clinicians would still need to sample some tissues at 100%? Can the authors comment on potential transfer learning or pretrained models?
> >
> > We appreciate the reviewer’s question regarding the practical reproducibility in new experimental settings. Below, we address this concern for two parts of our method: (i) our active sampling strategy SCRL, and (ii) the gene expression prediction model SCR²Net.
> >
> > 1. **Reproducibility of Our sampling strategy (SCRL):** The proposed SCRL sampler **can be initialized with any amount of existing ST data** and does **not** require full-coverage tissue sampling at the start. To assess the stability and reproducibility of the sampling strategy, we conducted repeated experiments using multiple random seeds (1, 42, 99, and 2026). The results of these additional experiments are summarized in *Table 2*, with detailed discussion provided in *Section 4.3*. Across different initializations, SCRL consistently achieves stable performance, demonstrating robustness to the choice of starting point. This behavior stems from the formulation of SCRL as a **sequential decision-making process**, where the sampler is optimized online based on the information observed so far and iteratively selects tissue locations to query until the predefined sampling budget is reached.
> >
> > 2. **Generalization of the prediction model (SCR$^2$Net):** For SCR$^2$Net, we acknowledge the role of pretrained models in enhancing generalization, as our vision encoder is initialized with pretrained weights. However, due to batch effects and inter-laboratory variations in tissue preparation and sequencing protocols, there typically exists a domain gap between training data and unseen experimental settings. Consequently, a small amount of data is required to fine-tune the model for new laboratory conditions, which is a common practice in ST modeling. Importantly, as more data become available in future studies, the model can be further optimized, which is expected to improve both the robustness and the transferability of the overall framework across different experimental settings.
> >
> > As suggested, we will added this discussion to the final version of manuscript in ***Appendix D (Practical Considerations and Future Directions)***.
> >
> > > ***Practical Considerations and Future Directions***
> >
> > > *From a practical perspective, the reproducibility and applicability of the proposed framework in new experimental settings can be considered from both the active sampling strategy (SCRL) and the gene expression prediction model (SCR$^2$Net) . The SCRL sampler is designed to operate with any amount of existing ST data and does not require full-coverage tissue sampling at initialization.  As summarized in Table 2 and discussed in Section 4.3, SCRL exhibits consistent and stable performance across different initializations, indicating robustness to the choice of starting point. This behavior is attributed to the formulation of SCRL as a sequential decision-making process, where the sampler is optimized online based on the information observed so far and iteratively selects tissue locations to query until the predefined sampling budget is reached.*
> >
> > > *For SCR$^2$Net, we leverage pretrained weights for the vision encoder to enhance representation learning and improve generalization across datasets. However, in real-world applications, batch effects and inter-laboratory variations in tissue preparation and sequencing protocols often introduce a domain gap between the training data and unseen experimental settings. Consequently, fine-tuning the model with a small amount of data from the target laboratory is typically required, which is a common and practical assumption in spatial transcriptomics modeling.*
> >
> > > *As more data become available in future studies, both the prediction model and the active sampling strategy can be further optimized, which is expected to improve the robustness, scalability, and transferability of the overall framework across diverse experimental conditions.*

---

> > ### Author Response · Authors · 2026-01-30
> > **Follow up Responses to Reviewer dCJ6 (2/2)**
> >
> > >Q6: Likewise, the answer to Q6 remains missing in the manuscript. The accuracy of the method is adequately reported, yet there is no warning or highlight regarding the impact that sampling errors could bring in diagnosis. Citing the author's answer to Q6: "The most challenging regions for both sampling and prediction are typically highly heterogeneous or transitional tissue areas, as well as regions containing rare cell populations". Rare cell populations are often critical in cancer diagnosis. Thus, it is convenient to clearly state these in the discussion of the results.
> >
> > We appreciate the reviewer’s suggestion on highlighting the diagnostic impact of sampling errors. As suggested, we will add this discussion to the final version of manuscript in ***Section 4.3 (Limitations and Challenges)***
> >
> > > ***Limitations and Challenges.** Despite these gains, overall accuracy remains bounded by intrinsic challenges of spatial transcriptomics rather than the sampling strategy alone. The most challenging regions for both sampling and prediction are typically highly heterogeneous or transitional tissue areas, as well as regions containing rare cell populations, which are often clinically important. In addition, morphologically similar regions may exhibit distinct molecular profiles, fundamentally limiting the predictive power of image-based models. Technical noise and spot-level signal averaging inherent to ST measurements further introduce uncertainty.*

---

### Author Rebuttal · Authors · 2026-01-24

**Rebuttal:**

We sincerely thank all the reviewers and ACs for their constructive feedback. This rebuttal addresses the common concerns raised by the reviewers (R1: dCJ6, R2: ACCL, R3: kU3P). We have modified the manuscript according to the reviews to address all of the concerns and comments proposed by the reviewers.

1. **Sensitivity analysis of reward design (R1, R2, R3):** We appreciate the reviewer’s comments on reward sensitivity. We added a sensitivity analysis of the reward weight configurations across different scales. The results of this newly added experiment are provided in *Figure 5* with a detailed discussion in *Section 4.3*.

   We observe thatgood and consistent performance is achieved only when different reward terms are placed at comparable scales, whereas configurations dominated by a single reward term or random sampling lead to clear performance degradation. These results demonstrate that the proposed reward design is effective without requiring fine-grained weight tuning, provided that a balanced contribution between biological diversity and spatial coverage is maintained.

2. **Computational costs (R1, R2, R3):** We thank the reviewer for the suggestion. We added a discussion of computational cost in the revised manuscript, with detailed statistics on model parameter size, training time, and sampling-time overhead summarized in *Table 4*, where the results show that the proposed method introduces only modest computational overhead.

3. **Visualization of sampling process (R1, R3):** To further illustrate the learning behavior of different sampling strategies, we add a visualization of the spatial distribution of selected spots on whole-slide images under a low sampling budget (10%) in *Figure 9*, with detailed explanation provided in *Appendix C*.

   The visualization reveals clear differences in sample selection patterns: entropy-based sampling tends to concentrate samples in limited regions, while diversity-based sampling spreads samples broadly but often allocates sampling budget to low tissue-density or background areas. In contrast, our method produces a more balanced sampling distribution by jointly considering spatial relationships and biological information, focusing on informative tissue regions while maintaining adequate spatial coverage.

**All modifications are highlighted in blue. Detailed point-by-point responses are provided below each reviewer’s comment.**

**Supporting Material:**

/attachment/ffe7d267c5963142748a14bb8dcad8adcbed97d2.pdf

---

### Meta-Review · Area_Chair_8e47 · 2026-02-01

**Recommendation:** Accept (Poster)
**Confidence:** 3

**Metareview:**

All the reviewers provided positive comments and scores for the work. The authors have also done a good job for rebuttal.

---

### Decision · Program_Chairs · 2026-02-13

Accept (Poster)